# Volumetric trans-scale imaging of massive quantity of heterogeneous cell populations in centimeter-wide tissue and embryo

Taro Ichimura[1]*, Taishi Kakizuka[1,2], Yoshitsugu Taniguchi[3], Satoshi Ejima[3], Yuki Sato[4], Keiko Itano[2], Kaoru Seiriki[5], Hitoshi Hashimoto[1,5], Ko Sugawara[6], Hiroya Itoga[6], Shuichi Onami[1,6], Takeharu Nagai[1,2,7]*

[1]Transdimensional Life Imaging Division, Institute for Open and Transdisciplinary Research, Initiatives, Osaka University, Osaka University, Osaka, Japan; [2]Department of Biomolecular Science and Engineering, SANKEN, Osaka University, Osaka, Japan; [3]SIGMAKOKI CO LTD, Midori, Sumida-ku, Tokyo, Japan; [4]Department of Anatomy and Cell Biology, Graduate School of Medical Sciences, Kyushu University, Fukuoka, Japan; [5]Laboratory of Molecular Neuropharmacology, Graduate School of Pharmaceutical Sciences, Osaka University, Osaka, Japan; [6]Laboratory for Developmental Dynamics, RIKEN Center for Biosystems Dynamics Research, Kobe, Japan; [7]Research Institute for Electronic Science, Hokkaido University, Sapporo, Japan

*For correspondence:
ichimura@otri.osaka-u.ac.jp (TI);
ng1@sanken.osaka-u.ac.jp (TN)

## eLife Assessment

The **important** study established a large-scale objective and integrated multiple optical microscopy systems to demonstrate their potential for long-term imaging of the developmental process. The **convincing** imaging data cover a wide range of biological applications, such as organoids, mouse brains, and quail embryos, but enhancing image quality can further enhance the method's effectiveness. This work will appeal to biologists and imaging technologists focused on long-term imaging of large fields.

**Abstract** We established a volumetric trans-scale imaging system with an ultra-large field-of-view (FOV) that enables simultaneous observation of millions of cellular dynamics in centimeter-wide three-dimensional (3D) tissues and embryos. Using a custom-made giant lens system with a magnification of ×2 and a numerical aperture (NA) of 0.25, and a CMOS camera with more than 100 megapixels, we built a trans-scale scope AMATERAS-2, and realized fluorescence imaging with a transverse spatial resolution of approximately 1.1 μm across an FOV of approximately 1.5×1.0 cm². The 3D resolving capability was realized through a combination of optical and computational sectioning techniques tailored for our low-power imaging system. We applied the imaging technique to 1.2 cm-wide section of mouse brain, and successfully observed various regions of the brain with sub-cellular resolution in a single FOV. We also performed time-lapse imaging of a 1-cm-wide vascular network during quail embryo development for over 24 hr, visualizing the movement of over $4.0×10^5$ vascular endothelial cells and quantitatively analyzing their dynamics. Our results demonstrate the potential of this technique in accelerating production of comprehensive reference maps of all cells in organisms and tissues, which contributes to understanding developmental processes,

brain functions, and pathogenesis of disease, as well as high-throughput quality check of tissues used for transplantation medicine.

## Introduction

Recently, life sciences have focused on comprehending the working principles of multicellular systems, spanning from basic biology to medical applications (*Sasai, 2013*; *McDole et al., 2018*; *Dominguez et al., 2023*; *Ueda et al., 2020*; *Rao et al., 2021*). Multicellular systems are generally complex, comprising heterogeneous cells rather than homogeneous assemblies. To understand the mechanism by which numerous cells cooperate to express the function of the entire system, it is ideal to observe all individual components within the system simultaneously during the time span of a whole phenomenon. In particular, in scenarios where a small fraction of cells in a large multicellular system may influence the fate of the population, or when distant cells and tissues operate synchronously, deducing the entire system from observations of sub-populations becomes challenging. Therefore, the trans-scale measurement of collective dynamics of all individual cells that constitute the system of interest is desired.

In response to this demand, researchers have recently reported the development of imaging methods with large field-of-view (FOV; *McConnell et al., 2016*; *Sofroniew et al., 2016*; *Fan et al., 2019*; *Ota et al., 2021*; *Yu et al., 2021*; *Ichimura et al., 2021*). Whereas most of these methods are focused on observing brain activity in neuroscience using two-photon excited fluorescence (*Sofroniew et al., 2016*; *Fan et al., 2019*; *Ota et al., 2021*; *Yu et al., 2021*), our efforts have been dedicated to developing techniques as versatile tools for studying multicellular systems science and developmental biology. In our previous work, we proposed a fluorescence imaging system for the visible wavelength region, capable of spatially resolving individual cells within a centimeter FOV, which we named AMATERAS-1 (A Multi-scale/modal Analytical Tool for Every Rare Activity in Singularity), enabling simultaneous observation of dynamics in the range of $10^5$–$10^6$ cells (*Ichimura et al., 2021*). Additionally, we

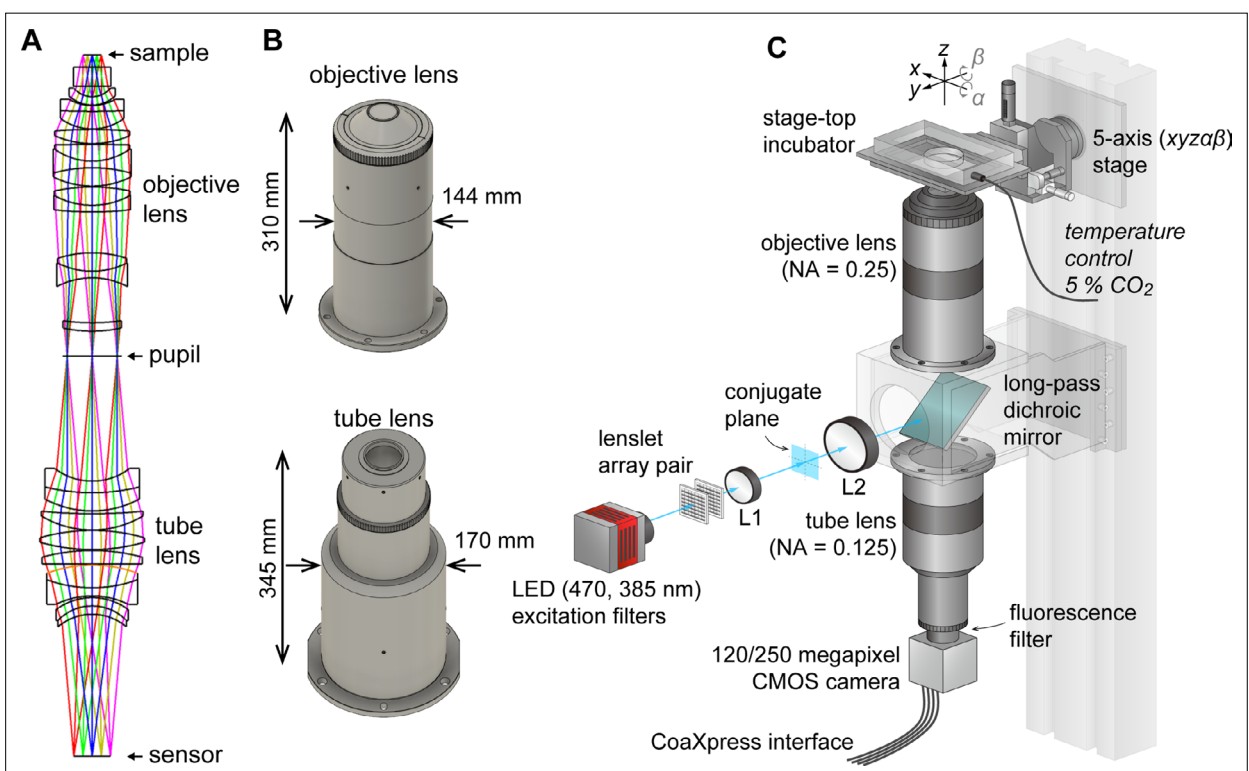

**Figure 1.** AMATERAS-2, a centimeter-FOV cell-imaging system. (**A**) Schematics of the design of the objective and tube lenses for AMATERAS-2. (**B**) Appearance of the objective and tube lenses with the width and length. (**C**) Diagram of the wide-field fluorescence imaging system, AMATERAS-2w, using the two lenses in (**A–B**). See Materials and methods for detail.

successfully detected less than 1% of cells with unique roles in multicellular systems and revealed how these rare cells trigger a drastic transition from unicellular to multicellular behavior in *Dictyostelium discoideum* (*Kakizuka et al., 2025*). However, the imaging system had a numerical aperture (NA) of 0.12, which is insufficient for realizing volumetric observation owing to its broad depth of field (DOF). Overcoming this limitation and enabling volumetric imaging for diverse three-dimensional (3D) tissues and small organisms has been a significant challenge.

In this study, we developed AMATERAS-2, a volumetric optical imaging system with approximately $15 \times 10$ mm$^2$ FOV, equivalent to that of AMATERAS-1 (*Ichimura et al., 2021*). The key component for the enhancement is the giant 2×lens system with an NA of 0.25, which offers improved 3D spatial resolution and higher sensitivity. By incorporating novel methodologies of optical sectioning and computational sectioning, we successfully added volumetric imaging capabilities. The effectiveness of these advancements was demonstrated in imaging 1.5-mm-thick brain section blocks and conducting a 25 hr time-lapse observation of vascular endothelial cells during quail embryogenesis, allowing us to trace the spatiotemporal dynamics of approximately $4.0 \times 10^5$ cells in three dimensions.

## Results
### Optical configuration and performance

AMATERAS-1 utilized a telecentric macro lens with ×2 magnification, thus allowing dynamic observation in the centimeter-FOV with subcellular resolution (*Ichimura et al., 2021*). Despite its efficacy in biological studies, its low NA (0.12) limits both its spatial resolution in the *z*-direction, and system brightness. To overcome these constraints and expand the possibilities for observing cell dynamics in tissues and embryos, we developed a giant lens system with a higher NA, consisting of a pair of objective and tube lenses. The objective lens is composed of 12 optical elements in 7 groups, while the tube lens is composed of 9 lenses in 6 groups (*Figure 1A*). Both lenses are infinity-corrected with NAs of 0.25 for the objective and 0.125 for the tube lens, and have a field number of 44 mm (diameter) suitable for large-area image sensors. They are effectively aberration-corrected to the diffraction limit or lower in the visible wavelength range of 436–656 nm. The objective lens and tube lens have sizes of 144 mm × 310 mm and 170 mm × 345 mm, respectively (*Figure 1B*). The objective and tube lenses exhibit zero vignetting, enabling observation of peripheral areas of the FOV with high brightness.

As the NA of AMATERAS-2 is 0.25, more than double that of AMATERAS-1 (0.12), our system in principle achieves over twice the spatial resolution in the transverse (*xy*) direction, more than four times the resolution in the longitudinal (*z*) direction, and over four times the brightness of AMATERAS-1.

Using this new lens system, we constructed a wide-field epi-illumination fluorescence imaging system, termed AMATERAS-2w, with a magnification of ×2, an NA of 0.25, and a field number of 44 mm (*Figure 1C*, Materials and methods). We utilized either a 120-megapixel CMOS camera or a 250-megapixel CMOS camera, depending on the research purpose. Both the cameras have an approximately 35 mm diagonal, thus providing an observation FOV of 17.8 mm and 17.4 mm diagonal at ×2 magnification, respectively. The pixel sizes for these cameras are 2.2 and 1.5 µm, with sampling intervals of 1.1 and 0.75 µm at ×2 magnification, respectively.

We installed a high-brightness LED light sources for fluorescence excitation, directing it into the objective lens through a custom-made long-pass dichroic mirror designed for GFP imaging with a cut-off wavelength of 497 nm. To ensure uniform illumination across the entire FOV, we employed a lens-let array pair. Additionally, we incorporated a 2-inch band-pass fluorescence filter between the tube lens and the camera.

We experimentally obtained the optical point spread function (PSF) using green fluorescent beads (center wavelength: 515 nm) with a 0.2 µm diameter, employing both the 120-megapixel CMOS camera (*Figure 2A*) and the 250-megapixel CMOS camera (*Figure 2B*). The left panels in *Figure 2A and B* display the PSFs in the *xy* and *xz* planes, respectively. The single-point spatial resolution was evaluated by the full-width of half-maximum (FWHM) of the line profiles determined through Gaussian-function fit. The transverse resolutions (*xy*-direction) obtained with the 120- and 250-megapixel cameras were $1.24 \pm 0.12$ (N=100) and $1.12 \pm 0.072$ (N=100; *Figure 2A–B*, right top), both of which are slightly downgraded from the theoretical FWHM, 1.05 µm, given by $0.51 \lambda_{em}/NA$ ($\lambda_{em}$em 515 nm, NA = 0.25; *Kino and Corle, 1996*; *Wilson, 2011*). This can be attributed to the coarse sampling interval (1.1 µm and 0.75 µm, respectively). In particular, in the 120-megapixel case, the apparent single-point resolution

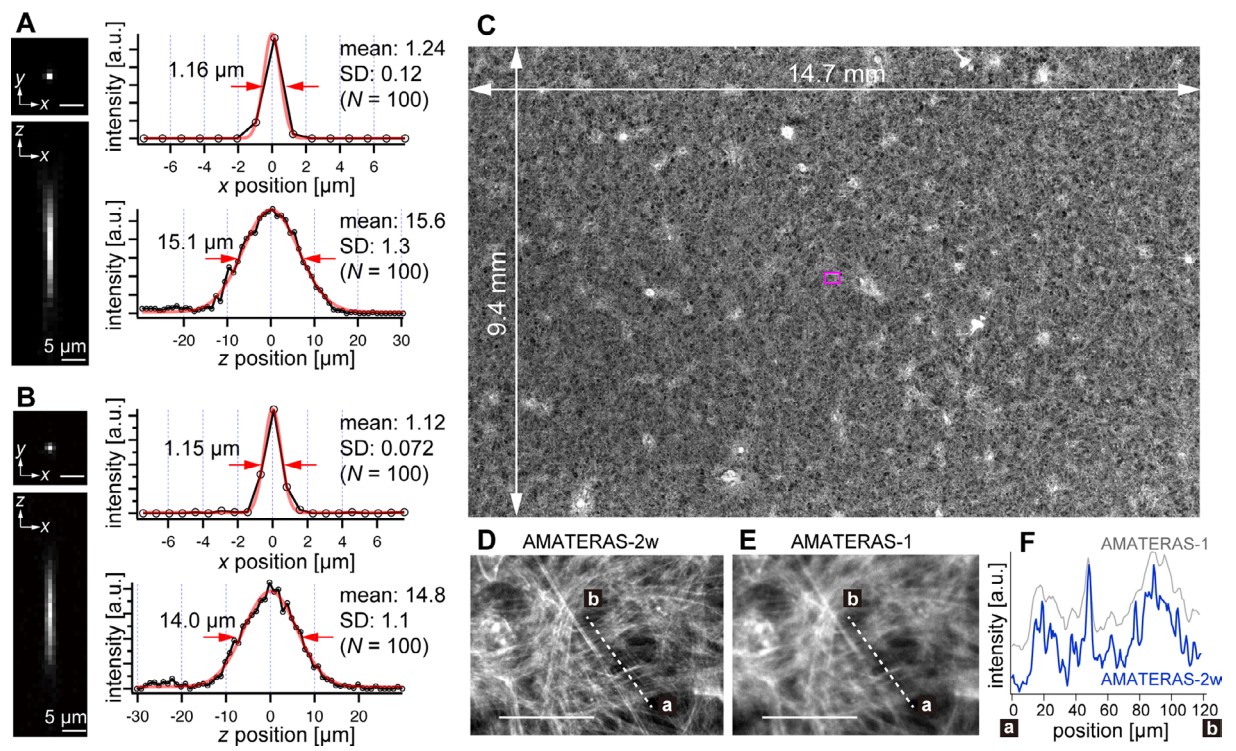

**Figure 2.** Optical performance of AMATERAS-2w. (**A**, **B**) Experimentally obtained PSF of green fluorescent beads with a 0.2 μm diameter in the *xy* and *xz* planes, captured using CMOS cameras with (**A**) 120 megapixels (pixel size=2.2 μm) and (**B**) 250 megapixels (pixel size=1.5 μm). The representative line profiles in the *x* and *z* directions on the beads are shown (black lines with circular markers) along with Gaussian fit curves (red lines) and FWHMs, and the mean and standard deviation (SD) calculated for 100 beads are described beside each profile. (**C**) Fluorescence image of a cardiomyocyte sheet in the entire FOV. (**D**) Digitally magnified view of the area indicated by a magenta square at the center of (**C**); scale bar, 100 μm. (**E**) Same area as (**D**) but observed by AMATERAS-1 for comparison. (**F**) Line profiles of the lines in (**D**) and (**E**) for comparison of spatial resolution.

The online version of this article includes the following figure supplement(s) for figure 2:

**Figure supplement 1.** Uncertainty in spatial resolution and position owing to undersampling.

varies depending on the relative position of the bead within the pixel (*Figure 2—figure supplement 1A–B*). The coarse sampling also introduces uncertainty in the estimated position of the bead's center coordinate (*Figure 2—figure supplement 1E*). Although this may pose some challenges in quantitatively evaluating fine shapes in cells, it is not a practical issue in imaging at cell resolution. By contrast, the 250-megapixel camera provides a nearly constant resolution (*Figure 2—figure supplement 1C–D*). Thus, the 250-megapixel camera is suitable when high-resolution images are required, whereas the 120-megapixel camera is suitable for other cases where increasing the number of photons per pixel or suppressing data size is necessary.

In the depth direction (*z*-direction), we evaluated the DOF by fitting Gaussian curves to the line profiles (*Figure 2A–B*, right bottom). The DOF was measured to be 15.6±1.3 (N=100) and 14.8±1.1 (N=100) for the 120- and 250-megapixel cameras, respectively. This difference in DOF is attributed to the distinct pixel sizes of the cameras (*Gross et al., 2008*). The DOF value for the 250-megapixel camera was found close to the theoretical FWHM, 14.3 μm, given by $1.78\,\lambda_{em}/NA^2$ ($\lambda_{em}$, em 515 nm, NA=0.25; *Wilson, 2011*).

The spatial resolution in the transverse direction has been significantly improved, approximately two times compared with that of AMATERAS-1. *Figure 2C and D* display fluorescence images of cardiomyocytes derived from human induced pluripotent stem cells (hiPSCs) stained with Rhodamine phalloidin (C) across the entire FOV and (D) within a central local region indicated by a magenta square in (C). These images were captured using the 250-megapixel camera, and the FOV size was 14.7×9.4 mm². For comparison, *Figure 2E* shows an image of the same area observed with AMATERAS-1 (*Ichimura et al., 2021*). Evidently, AMATERAS-2w delivers more detailed images with finer

spatial structures. This is further confirmed by the line profiles depicted in *Figure 2F*, where the filamentous actin stained with Rhodamine exhibits a sharper spatial resolution with AMATERAS-2w. This result indicated the capability of observing subcellular structures, such as myofibrils, in cell sheets with a large area, such as artificial myocardial sheets, which would enable us to simultaneously investigate microscale structures and macroscale multicellular dynamics.

## Spinning pinhole-array disk provides optical sectioning capability in wide-FOV

This imaging system utilizes wide-field illumination and detection, which currently lacks the capability to selectively acquire images of specific *z*-planes for volumetric imaging. To extend the system's applicability, we developed two methods to enable the volumetric imaging, namely, optical sectioning and computational sectioning.

For optical sectioning, various options of optical techniques exist for achieving 3D fluorescence imaging with a large FOV in the visible wavelength region. These include scanning confocal microscopy (*McConnell et al., 2016*; *Kino and Corle, 1996*), light-sheet microscopy (*Ueda et al., 2020*; *Battistella et al., 2022*), light-field microscopy (*Prevedel et al., 2014*), two-photon excitation microscopy (*Sofroniew et al., 2016*; *Fan et al., 2019*; *Ota et al., 2021*; *Yu et al., 2021*; *Demas et al., 2021*), and spatiotemporal focusing (*Papagiakoumou et al., 2020*). Among these options, we selected the confocal imaging method for this study. This is because the light-sheet microscopy struggles with uniform illumination over the centimeter scale FOV of AMATERAS (*Battistella et al., 2022*; *Voigt et al., 2019*), and the light-field microscopy involves a trade-off between spatial resolution and the ability to resolve three dimensions, which is incompatible with the AMATERAS's concept (*Nöbauer et al., 2023*). In addition, two-photon excitation microscopy, including that using spatiotemporal focusing, is crucial for deep tissue imaging, but still requires further innovations in pulsed laser power and scanning methods to be effectively applied to the wide FOV of AMATERAS.

We considered the confocal method to be the most compatible with our imaging system. Considering the vast FOV, we adopted the multipoint-scanning confocal system. The commonly used method involving a combination of a microlens array and a pinhole array disk (*Ichihara et al., 1996*; *Ishihara, 2003*) could not be employed in AMATERAS owing to the large NA of the tube lens (NA=0.125) compared with that of standard microscopes. Matching the NA requires shortening the distance between the two disks or enlarging the microlenses. However, shortening the inter-disk distance leads to a loss of space for inserting a dichroic mirror, and enlarging the microlenses reduces the number of foci. The confocal fluorescence microscopy requires a beam splitter due to the essential separation of fluorescence and excitation light. Therefore, designing microlens-pinhole arrays suitable for AMATERAS-2 presents a significant challenge. This challenge can be circumvented by using an optical configuration where excitation light and fluorescence pass through synchronized pinhole arrays on separate paths, as initially proposed in the early development of multipoint confocal microscopy (*Petráň et al., 1985*). However, this solution would require duplicating the optics of the giant lens, which is currently unfeasible. Moreover, even if possible, the added complexity of the system would render this approach impractical.

To address this challenge, we opted to use pinhole arrays without microlenses, which is rather a more traditional configuration of multipoint confocal microscopy (*Kino and Corle, 1996*; *Xiao et al., 1988*; *Halpern et al., 2022*). Although this method is less light-efficient compared with the one using microlenses, it aligns well with our imaging system's high NA and low magnification. Here, we designed the pinhole array disk for fluorescence imaging in green (e.g. GFP, YFP) and red (e.g. RFP, mCherry) wavelengths. Commercial confocal systems using pinhole-array disks typically feature large pinhole sizes (e.g. 50 µm) and wide spacing between pinholes (e.g. 200 µm), designed for general microscopes. For AMATERAS-2, however, with its ×2 magnification and NA of 0.25, the pinhole size must be much smaller. The diameters of the Airy disk for green and red wavelengths are about 5 µm and 6 µm, respectively, at the focal plane of the tube lens (NA=0.125). As a prototype, we set the pinhole size to 6 µm (*Figure 3A*), matching the Airy disk diameter for red wavelength. A more detailed discussion on the impact of pinhole diameter dependence on 3D resolution is available in *Figure 3—figure supplement 1* (see also Appendix File - Note 1). The pinhole spacing was set as 24 µm, striking a balance between throughput and crosstalk (*Figure 3A*). To implement this, we custom-made a pinhole-array disk and incorporated it into a rotary machine (CrestOptics, Rome, Italy).

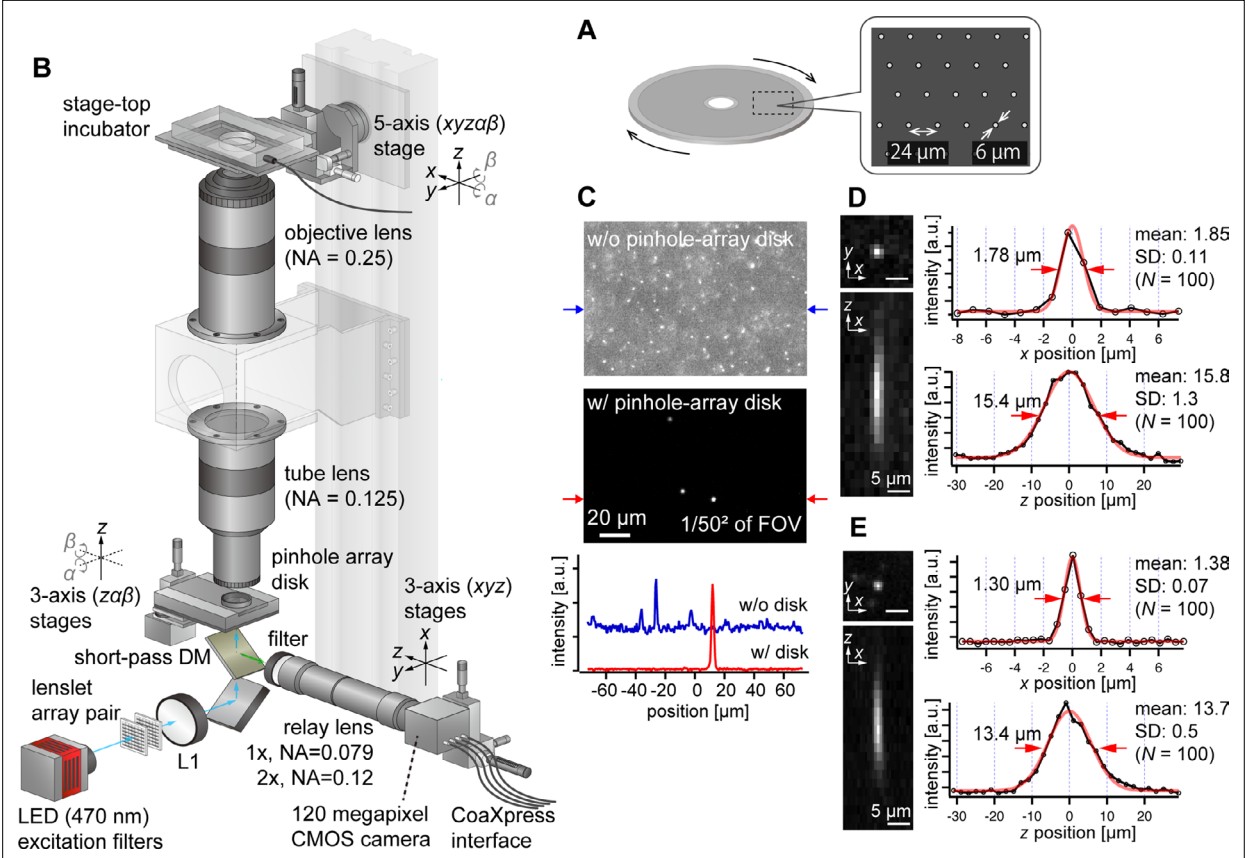

**Figure 3.** AMATERAS-2c, a multipoint confocal imaging system for 3D volume imaging. (**A**) Design of our pinhole-array disk. (**B**) Schematic of the optical configuration using the pinhole-array disk. (**C**) Effect of background light rejection by the presence of the pinhole-array disk in a 292×202 μm² area at the FOV center corresponding to 1/50² of the entire FOV. (**D**, **E**) Experimentally obtained PSF of green fluorescent beads with a diameter of 0.5 μm in the xy and xz planes, captured using relay lens with (**D**) ×1 magnification (NA=0.079) and (**E**) ×2 magnification (NA=0.12). The representative line profiles in the x and z directions on the beads are also shown (black lines with circular markers) along with Gaussian fit curves (red lines) and FWHMs, and the mean and standard deviation (SD) calculated for 100 beads are described beside each profile.

The online version of this article includes the following figure supplement(s) for figure 3:

**Figure supplement 1.** Schematic configuration of the confocal system and the effect of the pinhole size.

**Figure supplement 2.** Evaluation of the optical sectioning effect by the confocal system.

We positioned the pinhole array disk at the image plane of the tube lens and constructed a relay lens system to transfer the plane to the camera. This particular configuration, termed AMATERAS-2c, is illustrated in *Figure 3B* (Materials and methods). To split the light paths of fluorescence excitation light, we placed a short-pass dichroic mirror beneath the disk. For fluorescence excitation, we used a high-brightness LED with a center wavelength of 470 nm, the same as in the wide-field imaging system (*Figure 1C*). The relay lens consisted of a telecentric macro lens with ×1 magnification, and a 2-inch band-pass fluorescent filter was positioned at the entrance of the macro lens. If necessary, the relay lens can be replaced by a ×2 magnification lens to switch the total magnification from ×2 to ×4, based on resolution requirements.

Fluorescence images of 0.5 μm fluorescent beads were observed with and without the pinhole-array disk, where fluorescent beads were three-dimensionally dispersed in an agarose gel slab (*Figure 3C*). The presence of the pinhole-array disk dramatically suppressed background light from outside the focal plane. This verified that optical sectioning can be achieved by the pinhole-array disk. The variations in the 3D PSF with and without the pinhole-array disk are detailed in the *Figure 3—figure supplement 2*. Typical 3D PSF was obtained with the fluorescence beads with the relay lens of (D) ×1 and (E) ×2 magnifications (*Figure 3D and E*). As NA of the ×1 relay lens (NA=0.079) is lower than that of the tube lens (NA=0.125), the transverse spatial resolution (*Figure 3D*, right top) is degraded

compared to the one obtained by the wide-field configuration without the relay lens (*Figure 2A and B*, Appendix File - Note 1). By contrast, the ×2 relay lens (NA=0.12) almost matches with the tube lens, and the reduction in the transverse spatial resolution is relatively small (*Figure 3E*). As for the longitudinal resolution, the FWHM of the PSF was approximately 16 μm with a ×1 relay lens and approximately 14 μm with a ×2 relay lens, both values significantly exceeding the ideal value about 9.5 μm (Appendix File - Note 1). These discrepancies are attributed to the relatively loose focusing of the excitation light and the non-negligible size of the pinhole.

## Computational sectioning provides pseudo-depth-resolved imaging capability

In addition to optical sectioning, we developed a computational sectioning method to separate images of the focal plane and out-of-focal planes. During measurement, a *z*-stack is acquired in the wide-field imaging configuration (*Figure 1C*) by scanning the focal position in the *z*-direction and capturing images at each step. The raw image in a 3D volume results from the superposition of in-focus and out-of-focus plane images. We estimated the contribution from out-of-focus planes as

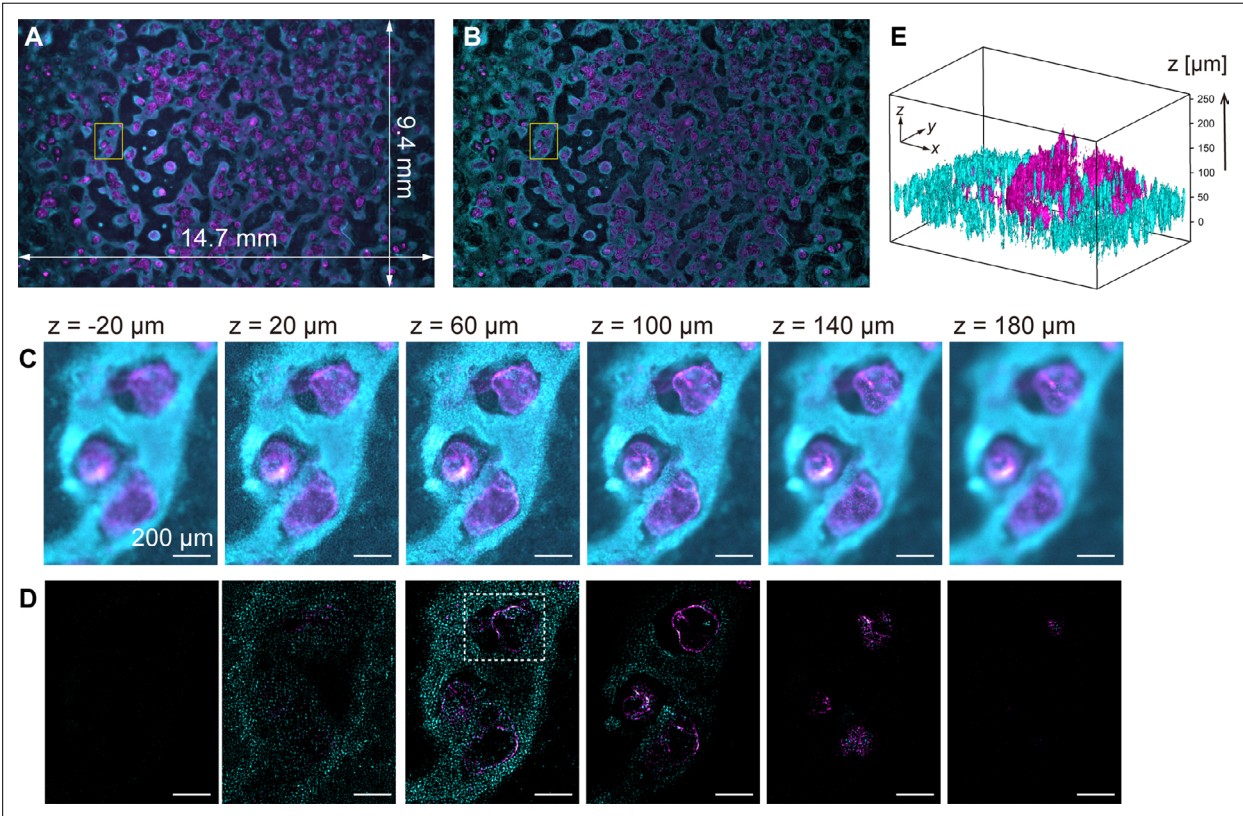

**Figure 4.** Volumetric imaging with computational sectioning. (**A**, **B**) Two-color images of the cavity chamber structure of a myocardial organoid (**A**) before and (**B**) after computational sectioning. Both (**A**) and (**B**) are MIP images of the *z*-stack data, captured using the 250-megapixel CMOS camera; here, magenta and cyan represent the distribution of immunostained cardiac troponin T and Hoechst-labeled nuclei, respectively. (**C**, **D**) Six z-layers with a difference of 40 μm in the yellow square region indicated in (**A**, **B**); scale bars, 200 μm. (**E**) Three-dimensional isosurface representation in the dashed square region indicated in (**D**).

The online version of this article includes the following video and figure supplement(s) for figure 4:

**Figure supplement 1.** Performance of computational sectioning with varied cutoff frequency.

**Figure supplement 2.** Dependence of baseline estimation performance on object size.

**Figure 4—video 1.** A *z*-stack of a cavity chamber structure of myocardial organoid (left) without and (right) with the computational sectioning, corresponding to *Figure 4C and D*, respectively.

https://elifesciences.org/articles/93633/figures#fig4video1

**Figure 4—video 2.** A 3D isosurface representation of a dome structure of cavity chamber with varying the view angle.

https://elifesciences.org/articles/93633/figures#fig4video2

the baseline by iteratively applying low-pass filtering under the assumption that the image's spatial frequency in the focal plane is higher than that in out-of-focus planes. The cut-off frequency of the low-pass filter was optimized using the principles of independent component analysis (Materials and methods, *Figure 4—figure supplements 1–2*, Appendix File – Note 2).

To demonstrate computational sectioning, we observed myocardial organoids derived from hiPSCs (*Ma et al., 2015*). This 3D dome structure of a cavity chamber is extensively studied in developmental biology and regenerative medicine as a model for human myocardial tissue development (*Hofbauer et al., 2021*; *Lewis-Israeli et al., 2021*). We fabricated a large area of organoids across the entire FOV, chemically fixed and immunostained them for cardiac troponin T (cTnT) with Alexa488, and stained the nuclei with Hoechst33342 (Materials and methods). The *z*-stack was obtained within a *z*-range involving the organoids (approximately 250 μm) with 4 μm intervals. *Figure 4A and B* display two-color overlaid images of the island-formed organoids across the entire FOV, without and with computational sectioning, respectively (both are maximum-intensity projection (MIP) images of the *z*-stacks). *Figure 4C and D* present images of six layers in the *z*-direction of the yellow square region of *Figure 4A and B*, before and after computational sectioning, respectively. After the sectioning, the distribution of cTnT and nuclei in each *z*-layer is clearly visualized (*Figure 4D*): The island areas are composed of multilayered cells, and the inter-island spaces are covered with a single layer of cells, as shown in *Figure 4D*, *z*=20 μm. A 3D isosurface representation (*Figure 4E*) shows that the hollow oval structure is formed by a thin layer of cTnT-positive cells (cardiomyocytes) whereas the underlying cell layer is composed of cTnT-negative cells, consistent with previous literature (*Ma et al., 2015*). This demonstration validates the technique for wide-field imaging. Video files of *Figure 4C–E* are available in *Figure 4—video 1* and *Figure 4—video 2*.

In principle, the signal component can be extracted as long as the signal intensity (in-focus component) is significantly higher than the statistical noise of the background intensity (out-of-focus component). This method is effective in cellular imaging when fluorescent molecules are localized within the cell, like in the nucleus, or when they exhibit a filament-like structure, thereby enabling cell recognition and dynamic tracking. It relies on the spatial frequency of the fluorescence image in the focal plane being uniform and clearly higher than that outside the focal plane.

## Observing a 1.2-cm wide and 1.5-mm-thick volume of mouse brain section

To demonstrate the potential for brain imaging, we performed imaging of a 1.5-mm-thick mouse brain section in the coronal plane (*Figure 5A*). The brain was chemically cleared using CUBIC (*Susaki et al., 2014*; *Tainaka et al., 2018*), and the cell nuclei were stained with SYTOX-Green (Materials and methods). The ×1 relay lens was primarily employed (total magnification=×2) to cover the coronal plane in the FOV. Each layer's exposure time was 1 s, and a total of 378 layers were acquired to cover the entire 1.5-mm-thick volume in steps of 4 μm in the *z*-direction. Optical and computational sectioning were both applied. *Figure 5B* shows a fluorescence image at a single *z*-layer (*z*=700 μm), revealing the 12 mm × 8 mm wide section with single-cell resolution. *Figure 5C* displays enlarged views of the *xy*-plane (coronal plane) in the dotted square region in *Figure 5B*, along with *xz* (transverse plane) and *yz* (sagittal plane) cross sections, essentially demonstrating successful *z*-direction sectioning. *Figure 5D* shows a 3D representation of the same volume of raw data obtained by optical sectioning alone, prior to computational sectioning. However, owing to intense background light overlapping, individual cells are challenging to distinguish, particularly in regions of high cell density. By utilizing computational sectioning to eliminate background light, clear images were obtained even in areas with strong background light. Image variation in the *z*-scan for the entire FOV and local regions can be seen in *Figure 5—video 1* and *Figure 5—video 2*.

*Figure 5C* visualizes the characteristic 3D structures of several regions, including hippocampal dentate gyrus, medial habenula, and choroid plexus, which are known to be associated with memory formation (*Hainmueller and Bartos, 2020*), depression (*Agetsuma et al., 2010*), and regulation of cerebrospinal fluid (*Dani et al., 2021*), respectively. Close-up views of the hippocampal region (indicated with the dashed square in *Figure 5B*) in the *xy* and *xz* planes are shown in *Figure 5E and F*, respectively. A comparison of the images at ×2 and ×4 magnification in *Figure 5E and F* reveals that the higher magnification (×4) allows for improved spatial resolution and separation of individual cells in both the *xy* and *xz* planes.

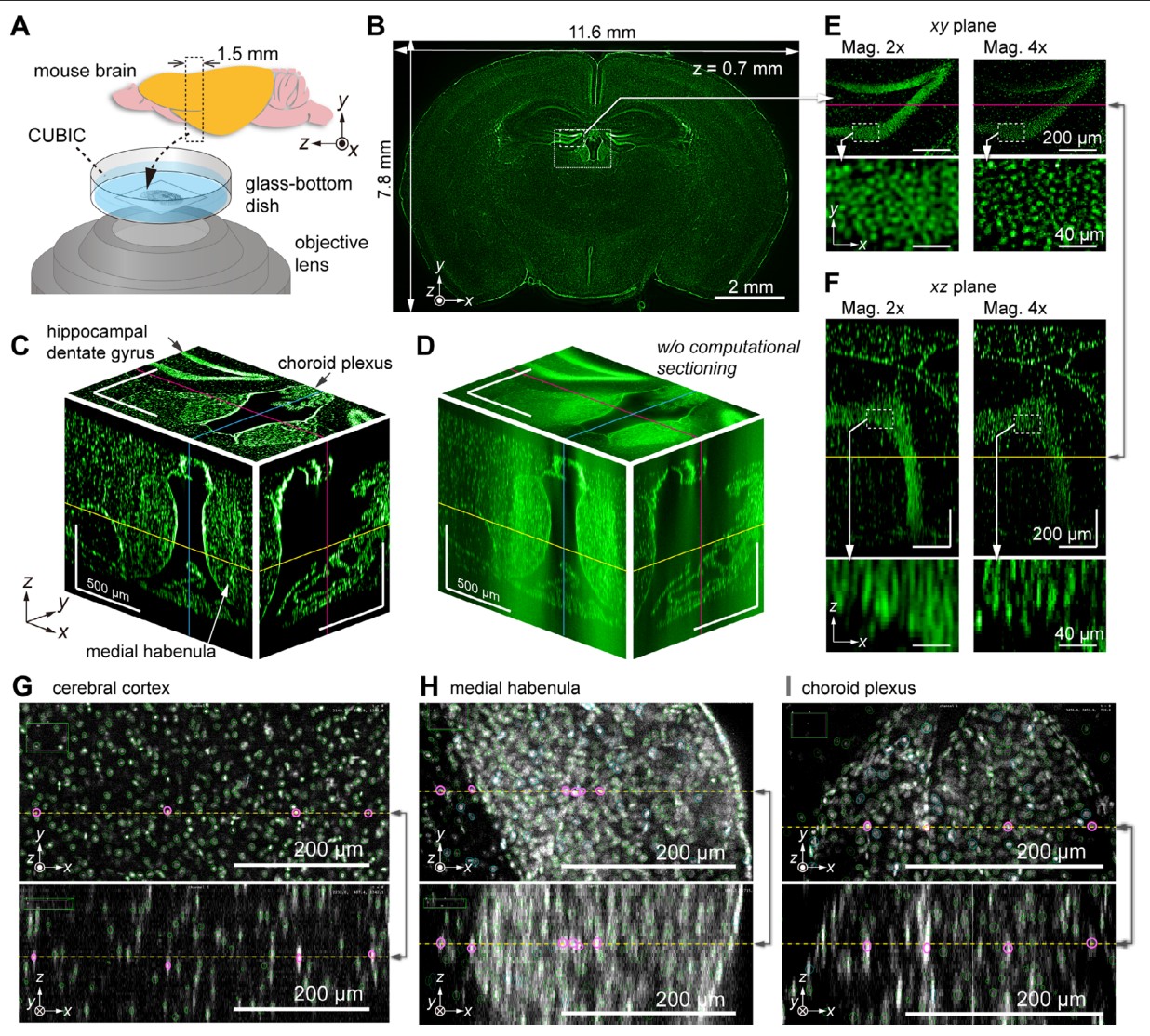

**Figure 5.** Volumetric imaging of a mouse brain section obtained by AMATERAS-2c. (**A**) Schematic of a mouse brain showing the placement of the brain sample in the inverted configuration. (**B**) Schematic of a single z-layer in the coronal plane with size of the effective area covering the entire brain section. (**C**) Fluorescence images of three orthogonal cross sections of the dotted square indicated in (**B**); here, the yellow, cyan, and magenta lines represent the z-, x-, and y-positions of the cross sections, respectively. (**D**) Cross-sectional images of raw data in the same region as (**C**), where the computational sectioning is not applied. (**E**, **F**) Magnified images of the hippocampal region (dashed square in (**B**)) in the (**E**) xy-plane and (**F**) xz-plane. For comparison, images obtained with ×2 (left) and ×4 (right) magnification systems are shown together. Note that the 3D regions observed at different magnifications do not perfectly match owing to the difficulty in optical alignment. (**G–I**) Cell detection by ELEPHANT in three brain regions, the cerebral cortex (**G**), medial habenula (**H**), and choroid plexus (**I**), where pairs of intersecting xy- and xz-planes are arranged vertically. The detected cells within each plane are marked with green and cyan ovals, and the straight line shared by the two planes is represented by a yellow dashed line, with certain cells along this line highlighted by magenta ovals. The xy-planes overlay the oval markers within a range of ±25 µm above and below the displayed planes, while the xz-planes show only oval markers on the planes.

The online version of this article includes the following video(s) for figure 5:

**Figure 5—video 1.** A z-stack of the mouse brain section shown in the full coronal plain region (*Figure 5B*).
https://elifesciences.org/articles/93633/figures#fig5video1

**Figure 5—video 2.** A z-stack of the mouse brain section shown in the local volume corresponding to *Figure 5C*.
https://elifesciences.org/articles/93633/figures#fig5video2

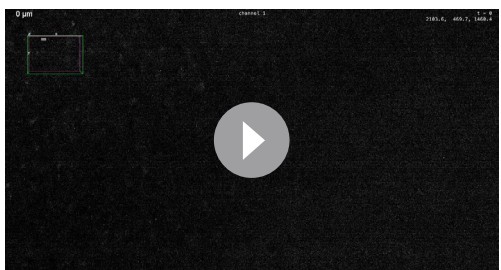

**Video 1.** A z-stack of the mouse brain section shown in the local volume in the cortex region. Cells detected by ELEPHANT are indicated by green oval markers. The z-position is swept with a step of 4 µm in 50 layers (200 µm) from the surface of the brain section.

https://elifesciences.org/articles/93633/figures#video1

For cell detection in the 3D imaging data of the mouse brain section, we used a recently developed interactive platform for cell detection and tracking, called ELEPHANT (Efficient LEarning using sParse Human Annotations for Nuclear Tracking; *Sugawara et al., 2022*). This platform seamlessly integrates manual annotation with deep learning and allows for result proofreading (Materials and methods). Using this advanced image analysis technique, cell nuclei were detected within the 3D volume. *Figure 5G* illustrates the cell detection results on specific xy and xz planes within the cortex. As the cell density in the cortex is relatively low, the ellipsoid-shaped nuclei image does not overlap with neighboring cells, resulting in the successful detection of nearly all cells (as clearly shown in the z-scan video, *Video 1*). The precision and recall of the cell detection were estimated to be 99.4% and 97.6 %, respectively (Materials and methods). Based on these results, the cell density in the cortex was calculated to be $1.18 \times 10^5$ cells/mm$^3$, which closely matches previous reports (*Herculano-Houzel and Lent, 2005*). This technique was also applied to two other brain regions, the medial habenula and choroid plexus (*Figure 5H–I*). In these regions, cell density is higher than in the cortex and the signal-to-noise ratio is lower due to strong background intensity, making it difficult for traditional methods to detect cells accurately. However, with the assistance of ELEPHANT, we successfully detected the majority of cells.

## Dynamics of over a $4.0 \times 10^5$ cells were captured over 24-hr development of a quail embryo

We employed the imaging methods described for a time-lapse observation of cell migration during quail development. The non-confocal optical configuration (AMATERAS-2w, *Figure 1C*) and utilizing computational sectioning alone were sufficient for this 250-µm-thick sample. We used a tie1:H2B-eYFP transgenic quail embryo (*Sato et al., 2010*) in which enhanced yellow fluorescent protein (eYFP) visualizes the nuclei of vascular endothelial cells (Materials and methods). The embryo was cultured *ex ovo* on a 35 mm glass-bottom dish (*Figure 6A*). Time-lapse fluorescence imaging began at 36 hr (HH10) after the start of egg incubation and captured eYFP signals in the developing embryo for 25 hr. For 3D imaging, we acquired z-stacks every 7.5 min, with each z-stack containing 21 layers spaced 12 µm apart. The intensity density of the excitation light was 113 mW/cm² at the sample plane, and the exposure time per layer was 1 s, under which conditions the quail embryo developed normally without significant phototoxicity or photobleaching. This time-lapse 3D imaging yielded a total of 4200 image layers (21 layers ×200 time points) with a data size of approximately 500 GB. During the time-lapse observation, we stabilized the relative distance between the lens and the sample using a self-developed autofocus system (Materials and methods).

Representative images of cell distribution based on the nuclear eYFP signals at four time-points (t=0, 8, 16, and 24 hr) are shown in *Figure 6B and C*. These images were reconstructed using the HSB color model to represent the 3D distribution after the baseline was removed by computational sectioning (*Figure 6—figure supplement 1*). The brightness indicates the intensity of the MIP image in the z-direction, whereas the hue represents the z-position with the maximum value in the MIP process (Materials and methods). As observed in *Figure 6B and C*, the cell nuclei distribution undergoes significant changes over time, thus resulting in the formation of organs such as the ventral aortae, heart, and dorsal aortae (t=24 hr, *Figure 6C*). The heart region appears blurred owing to its oscillating shape caused by beating. Time evolution of the HSB-color images in the entire FOV and magnified local regions can be seen in *Figure 6—videos 1–3*. *Figure 6D* shows enlarged views of the ventral aortae (indicated by the left dashed square in *Figure 6C*) at the four time-points. Additionally, a 3D isosurface of the z-stack of the dorsal aorta region (the right dashed square in *Figure 6C*) was calculated (*Figure 6E*, and *Figure 6—video 4*), clearly showing the developmental process of the two

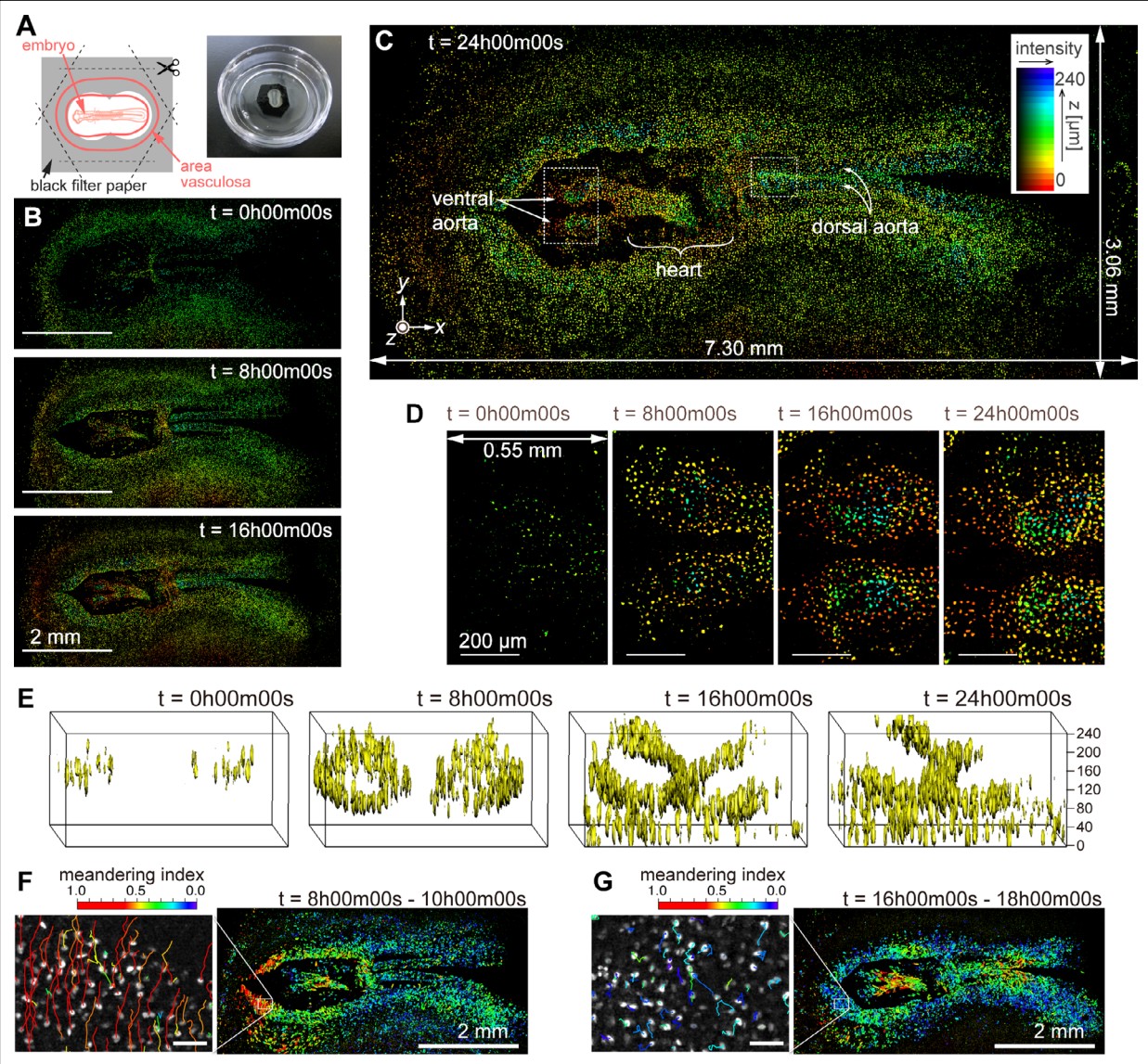

**Figure 6.** Dynamics of vascular endothelial cells in quail embryo captured by time-lapse observation. (**A**) Schematic showing how the specimen is mounted (left) and a photograph of a specimen placed in a glass-bottom dish. (**B**) Representative images obtained at $t$=0 hr, 8 hr, and 16 hr; scale bars, 2 mm. (**C**) Enlarged view of the image corresponding at $t$=24 hr; the pseudo-colors represent the $z$-position of the cells and fluorescence intensity by their hue and brightness, respectively. (**D**) Magnified views of the anterior side of ventral aortae (left dashed square in (**C**)) at the four time-points with intervals of 8 hr; scale bars, 200 µm. (**E**) Three-dimensional isosurface representation of the dorsal aorta region (right dashed square in (**C**)) viewed from the left side. (**F**), (**G**) Cell position trajectories and spatial mapping of features related to the cell dynamics in the two time-regions of (**F**) $t$=8–10 hr and (**G**) $t$=16–18 hr. Left panels show enlarged views in the white square regions (292×202 µm²) indicated in the right panels. In both the panels, the rainbow-colored trajectories of the cell movement are overlaid on the grayscale MIP images at the first frame in the time regions (F: $t$=8 hr, G: $t$=16 hr). The scale bars are 50 µm. The rainbow color table represents the meandering index of the cell movement in the two time-regions.

The online version of this article includes the following video and figure supplement(s) for figure 6:

**Figure supplement 1.** Computational processing and 3D visualization of the $z$-stack data of quail embryo.

**Figure 6—video 1.** Time lapse video of the quail development in: (6) the entire body region, (7) ventral aorta region, and (8) dorsal aorta region.
https://elifesciences.org/articles/93633/figures#fig6video1

**Figure 6—video 2.** Time lapse video of the quail development in: (6) the entire body region, (7) ventral aorta region, and (8) dorsal aorta region.
https://elifesciences.org/articles/93633/figures#fig6video2

**Figure 6—video 3.** Time lapse video of the quail development in: (6) the entire body region, (7) ventral aorta region, and (8) dorsal aorta region.
https://elifesciences.org/articles/93633/figures#fig6video3

*Figure 6 continued on next page*

*Figure 6 continued*

**Figure 6—video 4.** Time lapse video of the dorsal aorta region (*Figure 6E*) in the 3D isosurface representation.

https://elifesciences.org/articles/93633/figures#fig6video4

tube structures of the dorsal aortae (*Sato, 2013*). Note that the upper vessel wall is partially missing in the second half of the video because the fluorescent image was disturbed by the blood flow that began around $t$=15 hr.

At the current spatiotemporal resolution, we successfully traced the movement of individual cells. Segmentation of cell nuclei allowed us to detect approximately $4.5 \times 10^6$ cells across all 200 time points, and by linking cells in close proximity over consecutive time points, we tracked about $4.0 \times 10^5$ cells. *Figure 6F and G* show cellular movement along with a feature parameter related to cell movement at $t$=8 hr and 16 hr. Both left panels are magnified views of the square regions in the right panels, with trajectories drawn on MIP images. By analyzing the cell segmentation and tracking results, various feature parameters related to cell nucleus morphology and dynamics were computed, including nuclear size, brightness, aspect ratio, velocity, acceleration, mean square displacement, and meandering index (*Meijering et al., 2012*; *Beltman et al., 2009*). Here, we used the meandering index, a measure representing the straightness of cellular movement, to color the trajectories in *Figure 6F and G* with the rainbow color table. We found that the distribution of high meandering index at $t$=8 hr and 16 hr was significantly changed. Further detailed analysis and biological interpretation employing the cell-tracking results will be discussed in a future paper.

## Discussion

In this study, we presented AMATERAS-2w, a large FOV fluorescence imaging system, along with its variant AMATERAS-2c. The system is built around a newly designed giant lens with an NA of 0.25, magnification of ×2, and field number of 44 mm. This innovative lens configuration allows for imaging with an impressive spatial resolution of approximately 1.1 μm within a centimeter-scale FOV (15×10 mm²). Notably, this FOV-to-spatial resolution ratio surpasses those of previously reported large-FOV imaging methods. The key metric 'optical invariant', derived from the product of NA and FOV, has a value of 2.75 (Materials and methods, *Bumstead et al., 2018*), thus indicating excellent performance of the lens system. Furthermore, the 'space-bandwidth product', which signifies the FOV to spatial resolution ratio achieved under real measurement conditions (considering image sensor size and wavelength), reaches $4.3 \times 10^8$ (Materials and methods, *Lohmann et al., 1996*). Both of these metrics significantly outperform those reported in earlier studies on large FOV imaging methods (*McConnell et al., 2016*; *Sofroniew et al., 2016*; *Fan et al., 2019*; *Ota et al., 2021*; *Yu et al., 2021*; *Bumstead et al., 2018*). Although these metrics are essential indicators of our system's performance, they do not solely determine its superiority over other large-FOV imaging systems. Variations in biological targets and additional optical performance measures (such as frame rate and imaging depth) must also be considered. Our primary focus in designing this system was to expand the FOV, that is, maximize the number of observable cells at any instant, even at the cost of intracellular spatial resolution. Thus, in this aspect, our system excels as an instrument for such purposes.

The spatial resolution (FWHM) of AMATERAS-2w and –2 c was measured to be approximately 1.1–1.8 μm in the transverse direction (*xy*) and 13–16 μm in the longitudinal direction (*z*) through PSF measurements (*Figures 2A, B, 3D and E*). These values represent a significant enhancement compared with the previous AMATERAS version (2.3 and 64 μm, respectively; *Ichimura et al., 2021*). The improved transverse resolution enables clearer visualization of intracellular filamentous structures (*Figure 2F*). Moreover, the refinement in longitudinal resolution was more than four times greater, thus demonstrating that the expected effect was achieved with more than double NA. Depending on the object of observation, it may be debatable whether this anisotropic spatial resolution elongated along the optical axis qualifies 'cellular resolution'. Although this may be insufficient to resolve dense cell populations, it proves adequate for cells with only stained nuclei or those sparsely distributed, as observed in *Figures 5F–I , and 6E*. Using modern cell detection methods compatible with 3D big data, such as ELEPHANT employed in this study (*Sugawara et al., 2022*), we can achieve accurate cell detection, counting and tracking to some extent, even without clear spatial separation in three dimensions. It goes without saying that the higher spatial resolution in raw image data would enhance

the performance of subsequent image analysis. The next step in optical development involves further enhancing the NA of the lens system, including the relay lens, to further improve volumetric resolution and expand the range of observable biological objects.

We successfully achieved volumetric imaging of a 1.5-mm-thick mouse brain coronal section using the multipoint-scanning confocal imaging method, along with computational sectioning and tissue-clearing (*Figure 5*). This thickness corresponds to about one-tenth of the whole brain. The exposure time per image was 1 s, and data acquisition took less than 10 min. The ultimate goal is to apply this approach to enable whole-brain imaging. Unlike previous methods such as light-sheet imaging (*Susaki et al., 2014*) and block-face serial microscopy tomography (*Seiriki et al., 2019*), which require half a day or more for whole-brain data acquisition owing to tiling images from a narrow FOV, our method can significantly reduce the acquisition time when extended to the whole brain. This feature is effective not only for the brain imaging but also for imaging of other organs and whole body of an organism. We also plan to apply it to live imaging of highly transparent tissues such as zebrafish (*Takanezawa et al., 2021*) and organoids (*Hof et al., 2021*). Nevertheless, as the thickness increases, challenges such as increased background light and larger spherical aberration arise, thus resulting in weaker focal plane images. To address these issues and realize whole-brain imaging, we propose the incorporation of a mechanism for compensating spherical aberrations in the future. Additionally, we aim to improve the pinhole array pattern design and light source intensity to enhance transmittance and increase fluorescence intensity from samples, broadening the scope of applications.

We have shown that AMATERAS-2 can perform time-lapse observation of the centimeter-sized vascular networks in the quail embryo. It is a promising tool for understanding the multicellular behavior, especially the mechanisms of tissue formation during development (*Sasai, 2013*; *McDole et al., 2018*; *Dominguez et al., 2023*). In regard to the quail embryo's case, the data obtained here are the first demonstration of the dynamics of the extensive organization of the vascular network throughout the quail embryo at the single-cell level. Conventional methods of research employing traditional microscopy have selected a limited area for observation and thus had a difficulty in determining the distribution and synchronicity of cellular dynamics across the entire system. By analyzing such a huge four-dimensional data (*xyz* and *t*), we can gain insight into when and where each organ is formed, in what order, and in what interrelationships. In addition, employing this method allows us to break away from the conventional hypothesis-driven research manner and obtain new findings in a data-driven manner. This advantage can be realized not only for quail embryo but also for a variety of other species undergoing widespread and dynamic cellular events, making it a powerful tool in the study of multicellular organisms, especially in the study of tissue formation.

To further advance multicellular systems research, information on individual cell states is required in addition to cellular dynamics, therefore multiplex measurement is an essential technological challenge. Although most of the observations presented in this paper were made with monochromatic fluorescence imaging, multiplex imaging will lead to a more profound understanding if information on the distribution of cellular states can also be obtained by employing multiple fluorescent probes. This requires technological upgrades in optical filters and light sources for multicolor imaging. In addition, spatial transcriptomics has been rapidly developing in recent years and has become one of the most crucial tools for multicellular system study (*Rao et al., 2021*). In the near future, dynamics imaging and spatial transcriptomics for the same sample will be a very important technological integration to advance an integrative understanding of self-organization in tissue formation. Rather than simply using AMATERAS sequentially with existing spatial transcriptomics, it could be employed as an optical detection system for spatial transcriptomics to achieve extremely high throughput in the same wide FOV of cellular dynamics imaging.

Finally, let us discuss a practical challenge we encountered with our instrument, primarily related to the handling of extensive image data. In our studies, the raw data for 3D imaging of the mouse brain section and time-lapse 3D imaging of the quail embryo amounted to approximately 100 and 500 GB, respectively. However, after multiple image processing and analysis steps, a single experimental measurement resulted in several terabyte of image data. Managing such vast data proves challenging for standard computers in terms of both software and hardware capabilities. To address software limitations, we opted not to employ commercial software for data analysis. Instead, we utilized our originally developed programs, implementing certain optimizations to reduce analysis time, minimize write/read cycles, and prevent memory overflows. As for hardware, we established a data server with

petabyte storage, thereby enabling efficient data sharing among collaborators from diverse research institutions physically distant from one another. The existence of this infrastructure considerably facilitated the smooth progress of collaborative research within this study. Considering the increasing significance of handling image big data, we expect methodologies for image computation to become even more critical in the future. Ideally, a comprehensive system should be developed, encompassing not only data storage and sharing but also an integrated analysis solution in the cloud. We would like to actively promote the development of such a system, essentially encompassing advancements in imaging techniques and data-handling strategies.

# Materials and methods

### Key resources table

| Reagent type (species) or resource | Designation | Source or reference | Identifiers | Additional information |
|---|---|---|---|---|
| Genetic reagent (coturnix japonica) | tie1:H2B-eYFP | Sato et al., PLoS One. 5, e12674 (2010). | | |
| Strain, strain background (mouse) | C57BL/6JJmsSlc | Japan SLC,Inc. | RRID:MGI:5488963 | |
| Cell line (Homo sapiens) | iPS cell | Riken Cell Bank Takahashi et al., Cell 131, 861–872 (2007). | 201B7 RRID:CVCL_A324 | |
| Antibody | rabbit IgG anti-cTnT (polyclonal) | Bioss | Cat#:bs-10648R | 1:200 |
| Antibody | alexa488 conjugated goat IgG anti-rabbit IgG (polyclonal) | Abcam | Cat#:Ab150077; RRID:AB_2630356 | 1:500 |
| Other (culture medium) | CDM3 | Burridge, et al., Nat. Methods. 11, 855–860 (2014) | | used for initiation of iPSC differentiation |
| Other (culture medium) | StemFit | Ajinomoto | Cat#:AK02N | used for iPSC culture |
| Other (dish coating peptide) | iMatrix-511 | Matrixome | Cat#:89292 | used to coat a dish for iPSC culture |
| Other (cell detachment reagent) | TrypLE Select | Thermo Fisher Scientific, | Cat#:A1285901 | used to dissociate iPSCs |
| Chemical compound | Y-27632 | Fujifilm Wako | Cat#:030–24021 | |
| Chemical compound | CHIR99021 | Fujifilm Wako | CAS: 252917-06-9;Cat#:038–23101 | |
| Chemical compound | IWP-2 | Fujifilm Wako | CAS: 686770-61-6;Cat#:034–24301 | |
| Chemical compound | Medetomidine | Nippon Zenyaku Kogyo | CAS:86347-14-0 | |
| Chemical compound | midazolam | Sandoz Pharma | CAS:59467-70-8 | |
| Chemical compound | butorphanol | Meiji Seika Pharma | CAS:42408-82-2 | |
| Other (fluorescent dye) | Rhodamine phalloidin | Fujifilm Wako | Cat#:165–21641 | used for fluorescent labelling of actin filaments in the cardiac cell sheet |
| Other (fluorescent dye) | Hoechst33342 | Dojin | Cat#:H342 | used for fluorescent labelling of nuclei of cardiac organoid |
| Other (fluorescent dye) | SYTOX Green | Thermo Fisher Scientific | Cat#:S7020 | used for fluorescent labelling of nuclei in the mouse brain section |
| Other (tissue-clearing reagent) | CUBIC-L | Tokyo Chemical Industry | Cat#:T3740 | used for tissue clearing of the mouse brain section |
| Other (tissue-clearing reagent) | CUBIC-R+(M) | Tokyo Chemical Industry | Cat#:T3741 | used for tissue clearing of the mouse brain section |
| Software | ELEPHANT | *Sugawara et al., 2022*; *Sugawara, 2024* | Server: v0.5.7i, Client: v0.5.0 | https://github.com/elephant-track |

## Wide-field imaging system (AMATERAS-2w) configuration

We aimed for AMATERAS to achieve a larger FOV-to-resolution ratio than conventional microscopes, which required a giant imaging lens system. We realized this design concept by collaborating with SIGMAKOKI CO., LTD. Inc (Tokyo, Japan) to manufacture the giant objective and tube lenses as shown

in *Figure 1*. The characteristics of these lens are described in the main text. Designing such large lenses presents significant challenges in selecting optical glass materials compared to regular-sized lenses. Although there are over 200 types of optical glass available, our options are limited because the thickness or diameter of several types of glass may be unprocessable, or their transparency may degrade due to their size. Therefore, we iteratively explored the optimal design, considering the feasibility of material procurement and processing in conjunction with the optical design.

By combining these lenses, we constructed an imaging system with a 2×magnification, NA of 0.25, and a 44 mm field number. To facilitate both water-immersion and dry observation, we incorporated a glass plate whose thickness can be adjusted at the lens tip. The objective lens has a focal length of 120 mm, whereas its working distance is 14 mm owing to the presence of the attachment at the tip. The pupil positions of the objective lens and the tube lens are set externally to the lens bodies. This arrangement allows for the placement of a spatial filter or other devices at the pupil position, facilitating future enhancements in functionality.

For image acquisition, a 120-megapixel CMOS camera (VCC-120CXP1M, CIS, Tokyo, Japan) and a 250-megapixel CMOS camera (VCC-250CXP1M, CIS, Tokyo, Japan) were used, out of which we selected one depending on research purpose. Both the cameras have a chip size of 35 mm (diagonal). The pixel sizes of these two cameras are 2.2 µm and 1.5 µm, respectively. The sampling intervals are 1.1 µm and 0.75 µm at ×2 magnification. Image data captured by the CMOS cameras are loaded into a workstation via a CoaXpress frame grabber board (APX-3664G3, Aval Data, Tokyo, Japan). After the imaging experiments, they were transferred to a network server with large storage capacity for long-term storage and data sharing among project members.

By use of the lens system and CMOS cameras, a wide-field epi-illumination fluorescence imaging system was constructed. The imaging system chassis was designed to accommodate both inverted and upright microscope geometries. For this study, all experiments were conducted using the inverted microscope arrangement. Epi-illumination was achieved using a high-brightness LED sources (SOLIS-470C and SOLIS-385C, Thorlabs, Newton, NJ), with an excitation filter (#87–800, Edmund Optics, Barrington, NJ) placed immediately after it. The illumination light was homogenized using a pair of lenslet arrays and projected onto the sample surface. Additionally, a similar LED light source (SOLIS-525C, Thorlabs, Newton, NJ) with an illumination homogenizer was also mounted as transmitted illumination light for bright-field observation. To split the light path, we used a custom-made long-pass dichroic mirror measuring 158 mm × 120 mm×10 mm (BK7), which was designed for fluorescence imaging of GFP with a cut-off wavelength of 497 nm. A fluorescent filter (#86–992, Edmund Optics, Barrington, NJ) was positioned in front of the camera, which was firmly attached to the imaging lens using an F-mount.

The sample stage comprises a three-axis translational movable stage and a two-axis tilt stage. Only the *z*-axis stage is motorized, utilizing an electric actuator (SOM-C13E, SIGMAKOKI CO., LTD., Tokyo, Japan). For time-lapse observation under controlled conditions, we mounted a stage-top incubator (SV-141A, BLAST, Kawasaki, Japan) on the five-axis stage, providing a stable environment at 37 °C with $CO_2$ control. To ensure precise measurements, we enclosed the entire sample space within a dark box, shielding it from external light and temperature fluctuations.

## Multipoint confocal system (AMATERAS-2c) configuration

In the multipoint confocal imaging system, we utilized a custom-made pinhole array disk with specific dimensions (pinhole size: 6 µm, spacing: 24 µm). This disk was mounted on a high-speed rotating machine (CrestOptics, Rome, Italy) and placed precisely on the image plane of the imaging lens. Alignment throughout the FOV was ensured using a translation stage (XR25P/M, Thorlabs, Newton, NJ) and tilt stage (AIS-1016B, SIGMAKOKI CO., LTD., Tokyo, Japan) to adjust height and tilt. Beneath the disk, we positioned a short-pass dichroic mirror (DMSP490L, Edmund Optics, Barrington, NJ) to allow excitation light to be incident from below. The excitation light source included a high-brightness LED (SOLIS-470C, Thorlabs, Newton, NJ), an excitation filter (#87–800, Edmund Optics, Barrington, NJ), and a pair of lenslet arrays (#63–231, Edmund Optics, Barrington, NJ), together with a 75 mm lens (*f*=200 mm, LA1353-A, Thorlabs, Newton, NJ) to ensure uniform illumination onto the disk. The fluorescent image was reflected by the dichroic mirror, transmitted through a fluorescent filter (#86–992, Edmund Optics), and projected onto a 120-megapixel camera using either a 1×or 2×relay lens (LSTL10H-F and LSTL20H-F, Myutron, Tokyo, Japan). The relay lens and camera were attached with an

F-mount and secured on a three-axis translational stage (PT3/M, Thorlabs, Newton, NJ). To adjust the focus, the stage for the optical axis direction was motorized using an electric actuator (SOM-C25E, SIGMAKOKI CO., LTD., Tokyo, Japan).

## Measurement of 3D point spread functions

To evaluate the optical performance of the imaging systems, the 3D point spread function was measured using green fluorescent beads. Fluorescent beads with a 0.2 µm diameter (FluoSphere F8811, Invitrogen, Thermo Fisher Scientific, Waltham, MA) were employed in the evaluation of AMAT-ERAS-2w (*Figure 2A–B*) and those with a 0.5 µm diameter (FluoSphere F8813, Invitrogen, Thermo Fisher Scientific, Waltham, MA) were employed in the evaluation of AMATERAS-2c (*Figure 3D–E*). The beads were dispersed three-dimensionally in 1.5% (wt/vol) agarose gel on a glass-bottom dish, and *z*-stacks were obtained by moving the sample in the *z*-direction in 1 or 2 µm steps.

## Computational sectioning

In wide-field fluorescence imaging of a 3D volume, the superimposition of a fluorescent image from the focal plane and background light from outside the focal plane is a fundamental issue. This problem is not limited to wide-field imaging but also arises in multipoint confocal imaging of thick or high-density samples. To overcome this challenge, we employed computational sectioning, an image processing technique aimed at removing the background light component. Various algorithms have been proposed (*Lee and Yang, 2014*; *Walter and Ziesche, 2019*), and Leica microscopes' standard software includes this functionality. In this study, we developed an original algorithm as a simple and robust method for data analysis.

The critical step in our approach involves estimating the background light component at each layer of the *z*-stack data. To achieve this, we leverage the assumption that the background light exhibits a low spatial frequency, whereas the focal plane image demonstrates a high spatial frequency. Consequently, we employ an iterative low-pass filtering technique to estimate the background light component. Specifically, as shown in *Figure 6—figure supplement 1A and B*, a two-dimensional (2D) low-pass filter is applied to the raw data ($f_0(x,y)$) to obtain a smooth surface ($L_0(x,y)$). To obtain $f_1(x,y)$, $f_0(x,y)$ and $L_0(x,y)$ are compared and the smaller one is chosen to $f_1(x,y)$ at every position ($x,y$), as expressed by *Equation (1)*.

$$f_j\left(x,y\right) = \min\left(f_{j-1}\left(x,y\right), L_{j-1}\left(x,y\right)\right) \tag{1}$$

where min($a,b$) returns the smaller value of $a$ and $b$. Next, the low-pass filter is applied to $f_1(x,y)$ to obtain $L_1(x,y)$; this process is repeated to make $L_j(x,y)$ closer to the baseline of the original image. The iteration stops when the standard deviation of the difference of $f_j(x,y)$ and $L_j(x,y)$ reaches a preset value ($\varepsilon$).

$$\sqrt{Var\left(f_j\left(x,y\right) - L_j\left(x,y\right)\right)} < \varepsilon \tag{2}$$

Among various methods available as 2D low-pass filters, we sequentially applied one-dimensional (1D) infinite impulse response (IIR) filter in two directions ($x,y$). This method is faster than other methods such as a 2D Fourier transform (*Getreuer, 2013*). This speed difference is especially significant when the number of pixels in an image is massive, as in the present case. We adopted the Butterworth type of the IIR low-pass filter.

When the focal plane is empty and there is a bright object in the background, the baseline estimation tends to be skewed by the intensity fluctuations of the background. To mitigate this, we applied a 3×3 moving average filter to the entire image as a pre-processing step before initiating the baseline estimation algorithm. A critical parameter in our algorithm is the cutoff frequency of the low-pass IIR filter. If the cutoff frequency is set too high, the focal plane component would be included in the background; if it is set to low, background light would remain in the focal plane. Thus, it is crucial to find an optimal middle ground for the cutoff frequency. During this optimization, we did not account for cell size or optical system performance. Indeed, we employed a user-friendly blind separation method based on independent component analysis (ICA) to enhance usability (*Hyvaerinen et al., 2001*). Similar to ICA, we assumed that the fluorescence image in the focal plane deviates from the Gaussian (normal) distribution and that the superposition of images from multiple planes, including

both in-focus and out-of-focus planes, results in a distribution closer to the Gaussian distribution. To quantify the non-Gaussian nature of the distribution, we considered several measures, including kurtosis, skewness, negentropy, and mutual information (*Hyvaerinen et al., 2001*). Among these measures, we found skewness to be the most robust metric for our image dataset and incorporated it into our algorithm. The cutoff frequency was adjusted to maximize the skewness of the estimated in-focus image. The optimization of the cut-off frequency was performed on a subset of the data before applying it to the entire dataset. An example of the selection of non-Gaussianity measure and the optimization of the cut-off frequency is found in *Figure 4—figure supplements 1–2* (see also Appendix File - Note 2). The typical cut-off frequency was set at 0.06 in normalized frequency unit of 1/pixel, and the value was applied across all data.

## Culture of hiPSCs and differentiation into cardiac organoid with cavity chamber structure

We used a hiPSC line 201B7 (*Takahashi et al., 2007*), which was authenticated and confirmed mycoplasma negative by the supplier (RIKEN Cell Bank, Japan). The hiPSC line was routinely maintained as previously described (*Nakagawa et al., 2014*) on iMatrix-511 (Matrixome, 89292) coated culture dish in StemFit medium (Ajinomoto, AK02N), and 10 µM Y-27632 (Fujifilm Wako, 030–24021) was added for the first 24 hr after passage. Four days before inducing differentiation, hiPSCs were dissociated with TrypLE Select Enzyme (Thermo Fisher Scientific, A1285901) and $1.9 \times 10^5$ cells were passed into iMatrix-511 coated 12 well plate in StemFit medium supplemented with 10 µM Y-27632. After 24 hr, the medium was changed to StemFit without Y-27632 and was exchanged daily for 3 days. Differentiation was initiated with the CDM3 medium (*Burridge et al., 2014*) composed of RPMI-1640 with HEPES (Fujifilm Wako, 189–02145), 0.5 mg/ml human recombinant albumin (Sigma Aldrich, A9731) and 0.2 mg/ml L-ascorbic acid 2-phosphate (Sigma-Aldrich, A8960) supplemented with 3 µM CHIR99021 (Fujifilm Wako, 038–23101) for 48 hr. On day 2 of differentiation, the medium was changed to CDM3 supplemented with 5 µM IWP-2 (Fujifilm Wako, 034–24301). From day 4 to day 8, the cells were cultured in CDM3 and the medium was changed every other day. From day 8, the medium was changed to RPMI-1640 with HEPES supplemented with 2% B27 (Thermo Fisher Scientific, 17504044) and was changed every other day. The cells were fixed in 4% paraformaldehyde (PFA; Thermo Fisher Scientific, 43368) on day 15 for immunocytochemical analysis.

## Immunocytochemical staining of cardiac organoid

Cells were fixed with 4% PFA for 15 min at room temperature (RT ~23 °C), permeabilized with 0.2% Triton X-100 (Fujifilm Wako, 807423) for 20 min at RT and incubated with blocking solution composed of D-PBS with 5% BSA and 10% FBS for 30 min at RT. Subsequently, the cells were incubated with primary antibody (1:200 rabbit IgG anti-cTnT, Bioss, bs-10648R) at 4 °C overnight. After washing three times, cells were incubated with secondary antibody (1:500 alexa488 conjugated goat IgG anti-rabbit IgG, Abcam, ab150077) for 2 hr at RT. The cells were washed three times in D-PBS with 1 µg/ml Hoechst33342 (Dojin, H342). Subsequently, the cells were incubated with D-PBS and used for fluorescent observation.

## Color representation of the 3D image

*Figure 6B–D* show the 3D positions of the cells using the HSB (also known as HSV) color model. After applying the method described earlier to remove background light from the *z*-stack image, we conducted a maximum value intensity projection along the *z*-direction. The *z*-position of the maximum value at each *xy*-position was then recorded (*Figure 6—figure supplement 1C*). The color image was reconstructed by associating the brightness and hue of the HSB table with the value of the maximum intensity and the maximum value position, respectively (*Figure 6—figure supplement 1D*). For the hue table, we assigned red to blue to *z*-layer numbers ranging from 0 (*z*=0 µm) to 20 (*z*=240 µm).

## Preparation of mouse brain section

All animal care and handling procedures were conducted with approval from the Animal Care and Use Committee of Osaka University (Authorization number: R02-8-7). Our utmost efforts were made to minimize the number of animals used. Experiments involved adult male C57BL/6J mice (SLC, Shizuoka, Japan) aged between 2 and 3 mo. To ensure proper anesthesia, mice were deeply

anesthetized through intraperitoneal injection of an anesthetic cocktail containing 0.3 mg/kg mede-tomidine (Nippon Zenyaku Kogyo, Fukushima, Japan), 4 mg/kg midazolam (Sandoz Pharma, Basel, Switzerland), and 5 mg/kg butorphanol (Meiji Seika Pharma, Tokyo, Japan). Subsequently, anesthetized mice underwent transcardial perfusion with saline, followed by 4% PFA (Nacalai Tesque, Kyoto, Japan) dissolved in phosphate-buffered saline (PBS). Brain tissues were then excised and immersed in a 4% PFA solution until further use. For tissue preparation, brain tissue blocks were sliced into 1.5-mm-thick coronal sections using a vibrating microtome (LinearSlicer Pro7N, Dosaka EM, Kyoto, Japan). Subsequent staining and tissue clearing involved immersion of sections in CUBIC-L (Tokyo Chemical Industry, Tokyo, Japan) for delipidation, SYTOX Green (Thermo Fisher Scientific, Waltham, MA, USA) solution (1:5000) diluted in 20% (vol/vol) DMSO in PBS for nuclear staining, and CUBIC-R+(M) (Tokyo Chemical Industry, Tokyo, Japan) for refractive index matching, following previously established methods (*Tainaka et al., 2018*; *Susaki et al., 2020*). After clearing, tissue sections were placed on glass bottom dishes and embedded in 2% (wt/vol) agarose gel prepared with CUBIC-R+(M) for subsequent imaging analysis.

## Preparation of quail embryo

Transgenic quail line, tie1:H2B-eYFP (*Sato et al., 2010*) was bred in quail breeding facility at Kyushu University. Fertilized eggs were incubated at 38 °C. The staging of quail embryos was based on the Hamburger and Hamilton stages of chicken embryos (*Hamburger and Hamilton, 1951*). The animal study was approved by the Institutional Animal Care and Use Committee of Kyushu University (Authorization number: A20-019). Ex ovo culture was performed as previously described (*Sato and Lansford, 2013*). Black filter paper was used instead of white filter paper to avoid fluorescence background.

## Autofocus in time-lapse imaging

To address *z*-directional drift in the time-lapse observation of quail, we developed an original autofocus method. During the intervals between fluorescence *z*-images, we acquired bright-field *z*-stacks and analyzed the images to identify the most in-focus position. To achieve this, we used a black ink marker located on the substrate edge as the autofocus target instead of the variable sample itself. For evaluating the degree of in-focus, we applied a total variation (TV) filter to the marker image and used the kurtosis of the filtered image as the focus score. The *z*-position with the maximum focus score was identified as the in-focus location, thus signifying the sharpest image of the marker edge. Although other filter types and statistical moments are available options for the focus score (*Mateos-Pérez et al., 2012*), our preliminary experiments demonstrated the suitability of the kurtosis of TV for the marker image. In the actual experiment, we acquired 15 images in 15 μm steps and plotted the focus scores against *z*-positions. Through quadratic function fitting of the focus scores at three *z*-positions (including the *z*-position with the maximum value and adjacent positions), we estimated the best in-focus position with an accuracy of 1 μm.

## Deep-learning based cell detection for the mouse brain data

Cell detection was conducted using ELEPHANT, a unified platform that facilitates manual annotation, deep learning and proofreading of results within a single user-friendly GUI (*Sugawara et al., 2022*; *Sugawara, 2024*). ELEPHANT serves as an extension to Mastodon (*Tinevez et al., 2025*), an open-source framework for large-scale tracking deployed in Fiji, a widely used image-analysis software (*Schindelin et al., 2012*). ELEPHANT supports incremental deep learning with sparse annotation, incorporating algorithms for detecting cells in 3D and tracking them in time-lapse 3D image datasets. In this paper, we utilized the cell detection function exclusively for analyzing 3D imaging data of mouse brain section (*Figure 5G–I*). The incremental learning approach allows cell detection models to be trained progressively on a dataset that begins with sparse annotations and is continuously enhanced through human proofreading. For visualization, image data were displayed using BigDataViewer (*Pietzsch et al., 2015*) on Mastodon, accessed via BigDataServer (*Pietzsch, 2024*). This setup permits the visualization of large-scale image data on client computers while maintaining the data on the server. Deep learning capabilities were provided by the ELEPHANT server, enabling remote GPU access. This setup allows non-experts to perform image analysis on big data using deep learning, effectively overcoming typical challenges such as the need for large amounts of high-quality training data, the absence of an interactive user interface, and limited access to computing resources,

including substantial storage and high-end GPUs (*Moen et al., 2019*; *Wen et al., 2021*). The server computer used in this study is equipped with an Intel(R) Xeon(R) Gold 6132 CPU @ 2.60 GHz, runs Ubuntu 20.04, and includes 384 GB DDR4 2,666 MT/s RAM, 2 x NVIDIA Tesla V100-PCIE-32GB GPUs, and a network-attached Lustre parallel file system with over 2 PB of storage. The client computer features an Apple M1 Pro CPU, runs Sonoma 14.3, and has 16 GB of LPDDR5-6400 RAM and a 500 GB SSD.

In the analyses presented in this paper (*Figure 5G–I*), cell detection models were trained for three regions, including the cortex. In each region, the detection model was refined by repeating the sequence (annotation, training, prediction, and proofreading) five to seven times. By annotating approximately 100 cells in total, we were able to establish a model capable of detecting the vast majority of the cells. The training of the detection models was conducted on volumes of 256×256 x 16 voxels, which were prepared by preprocessing with a random flip in each dimension. During the label generation step, the center ratio was set to 0.4 and the background threshold was set to zero, meaning that only explicitly annotated voxels were used for training. In the prediction step, volumes cropped to dimensions of 700×700 × 100 around the target area were used as input. In the postprocessing step, a threshold for nucleus center probabilities was set to 0.5, and $r_{min}$, $r_{max}$ and $d_{sup}$ were set to 1 μm, 5 μm, and 3 μm, respectively (see details about the parameters in *Sugawara et al., 2022*).

To quantitatively evaluate the performance of cell detection, we calculated precision and recall, defined as TP/(TP +FP) and TP/(TP +FN), respectively, where TP stands for true-positive, FP for false-positive, and FN for false-negative. In the actual 3D data of the cortical region, the values of TP, FP, and FN were manually counted by comparing the detection results of cells (N~640) within a defined volume of 310×170 × 100, one by one, with the actual images. As a result, we obtained a precision of 99.4% and a recall of 97.6%.

## Cell segmentation and tracking for the quail embryo data

For the analysis of quail embryo data, cell detection, segmentation and tracking were performed using a custom-made program coded in Python. The observed 3D volume has a transverse area (*xy*) of more than 1 cm$^2$ and a height (*z*) of 240 μm, which is relatively thin in the *z* direction compared to the *xy* plane. The overlap of multiple vascular endothelial cells in the *z* direction occurs with a very low probability. Therefore, to save computational costs and time, 2D cell detection was conducted on *z*-projected MIP images instead of full 3D cell detection. To avoid miss-detection of overlapping cells, we divided the *z*-stack data into three blocks along the z-axis and created MIP images for each block, on which 2D cell detection was performed on the MIP images. The resulting cell lists were compared to identify identical cells detected across multiple blocks. By recognizing double-detected cell pairs as identical, we updated the cell list accordingly. This process was applied to all double-detected cells, thus achieving effective cell detection in the 3D volume and enabling cell tracking throughout the observation period.

## Evaluation of optical system with optical invariant and space-bandwidth product

We employed two indices to evaluate the lens system's scale range and compare it with previous research. The first index, the optical invariant, measures the lens system's performance and is obtained by multiplying the FOV radius and NA. The second index, the space-bandwidth product, considers the image sensor and wavelength while quantifying the ratio of actual resolution to FOV. These indices are calculated using the following simplified formulas, respectively (*Bumstead et al., 2018*).

$$I = \frac{NA \times FN}{2M} \tag{3}$$

$$SBP = 8 \left( \frac{R_{FOV}}{d_{xy}} \right)^2 \tag{4}$$

where *I*, *FN*, *M*, *SBP*, $R_{FOV}$, and *dxy* denote optical invariant, field number, magnification, space-bandwidth product, FOV radius, and spatial resolution (FWHM).

## Acknowledgements

We would like to thank Prof. K Fujita of Department of Applied Physics, Osaka University, Japan for his valuable comments on optics design. We are also grateful to Prof. S Miyagawa, and Prof. Y Sawa of Graduate School of Medicine, Osaka University, for their support on handling human iPS cells and valuable discussion. Funding: Grant-in-Aid for Scientific Research on Innovative Areas "Singularity Biology (No. 8007)" 21H00431 (YS), 18H05416 (HH), 18H05412 (SO), and 18H05410, 18H05408 (TN). Grant-in-Aid for Transformative Research Areas (A) "Seeing through Scattering Media (No. 20A207)" 21H05590 and 23H041350 (TI) the Research Program of Five-star Alliance in NJRC Mater. & Dev. (TN) Precursory Research for Embryonic Science and Technology (PRESTO)JPMJPR18G2 (TI), JSPS KAKENHI JP23H00395 and JP24K22022 (HH), AMED Brain/MINDS, JP21dm0207117 (HH) and BINDS JP23ama121054 and JP23ama121052 (HH), The Uehara Memorial Foundation (TN), Takeda Science Foundation (TN, HH), Core Research for Evolutionary Science and Technology (CREST) JPMJCR15N3 (TN), JPMJCR1926 (KS, SO), RIKEN Cluster for Science, Technology and Innovation Hub (SO), JST NBDC Grant Number JPMJND2201 (SO).

## Additional information

### Competing interests

Yoshitsugu Taniguchi: Y.T. is an employee of SIGMAKOKI CO., LTD. Satoshi Ejima: S.E. is an employee of SIGMAKOKI CO., LTD. Ko Sugawara: K.S. is employed part-time by LPIXEL Inc. The other authors declare that no competing interests exist.

### Funding

| Funder | Grant reference number | Author |
|---|---|---|
| Japan Society for the Promotion of Science | 21H00431 | Yuki Sato |
| Japan Society for the Promotion of Science | 18H05416 | Hitoshi Hashimoto |
| Japan Society for the Promotion of Science | 18H05412 | Shuichi Onami |
| Japan Society for the Promotion of Science | 18H05410 | Takeharu Nagai |
| Japan Society for the Promotion of Science | 18H05408 | Takeharu Nagai |
| Japan Society for the Promotion of Science | 21H05590 | Taro Ichimura |
| Japan Society for the Promotion of Science | 23H041350 | Taro Ichimura |
| Japan Society for the Promotion of Science | JP23H00395 | Hitoshi Hashimoto |
| Japan Society for the Promotion of Science | JP24K22022 | Hitoshi Hashimoto |
| Japan Science and Technology Agency | 10.52926/JPMJPR18G2 | Taro Ichimura |
| Japan Agency for Medical Research and Development | JP21dm0207117 | Hitoshi Hashimoto |
| Japan Agency for Medical Research and Development | JP23ama121054 | Hitoshi Hashimoto |
| Japan Agency for Medical Research and Development | JP23ama121052 | Hitoshi Hashimoto |

| Funder | Grant reference number | Author |
|---|---|---|
| Japan Science and Technology Agency | 10.52926/jpmjcr15n3 | Takeharu Nagai |
| Japan Science and Technology Agency | 10.52926/jpmjcr1926 | Ko Sugawara Shuichi Onami |
| Japan Science and Technology Agency | 10.52926/jpmjnd2201 | Shuichi Onami |
| Uehara Memorial Foundation | | Takeharu Nagai |
| Takeda Science Foundation | | Hitoshi Hashimoto Takeharu Nagai |

The funders had no role in study design, data collection and interpretation, or the decision to submit the work for publication.

## Author contributions

Taro Ichimura, Conceptualization, Resources, Data curation, Software, Formal analysis, Funding acquisition, Validation, Investigation, Visualization, Methodology, Writing – original draft, Writing – review and editing; Taishi Kakizuka, Resources, Data curation, Visualization, Methodology; Yoshitsugu Taniguchi, Satoshi Ejima, Resources, Methodology; Yuki Sato, Resources, Data curation, Funding acquisition, Methodology; Keiko Itano, Data curation, Software, Visualization; Kaoru Seiriki, Resources; Hitoshi Hashimoto, Resources, Funding acquisition, Methodology; Ko Sugawara, Data curation, Software, Funding acquisition, Validation, Visualization; Hiroya Itoga, Data curation, Software; Shuichi Onami, Data curation, Software, Funding acquisition; Takeharu Nagai, Conceptualization, Funding acquisition, Methodology, Project administration, Writing – review and editing

## Author ORCIDs

Taro Ichimura ⓘ https://orcid.org/0000-0002-3740-3634
Hitoshi Hashimoto ⓘ https://orcid.org/0000-0001-6548-4016
Shuichi Onami ⓘ https://orcid.org/0000-0002-8255-1724

## Ethics

All animal care and handling procedures were conducted with approval by the Institutional Animal Care and Use Committee of Kyushu University (Authorization number: A20-019) and Osaka University (Authorization number: R02-8-7).

Reviewer #1 (Public review): https://doi.org/10.7554/eLife.93633.3.sa1
Reviewer #2 (Public review): https://doi.org/10.7554/eLife.93633.3.sa2
Author response https://doi.org/10.7554/eLife.93633.3.sa3

# Additional files

## Supplementary files

MDAR checklist

## Data availability

All bioimaging data generated in this study are present in the paper and/or Supplementary Materials. The image datasets are available at ssbd-repos-000409, https://doi.org/10.24631/ssbd.repos.2024.12.409 in SSBD:repository.

The following dataset was generated:

| Author(s) | Year | Dataset title | Dataset URL | Database and Identifier |
|---|---|---|---|---|
| Ichimura T, Nagai T, Kakizuka T | 2024 | A set of image data used for demonstration of a newly developed optical imaging system "AMATERAS-2" | https://doi.org/10.24631/ssbd.repos.2024.12.409 | SSBD:repository, 10.24631/ssbd.repos.2024.12.409 |

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

## Appendix 1

### Note 1. Discussion on the confocal imaging system: Optical configuration, pinhole size and spatial resolution

This section describes the details of the confocal system configuration and the setting of the pinhole size. AMATERAS in this paper is designed for fluorescence imaging in the visible wavelength region. Currently, the system is a single-channel observation system using excitation light at 470–490 nm and observing fluorescence in 510–540 nm region, which is primarily targeted at green-to-yellow fluorescent proteins such as EGFP (emission peak: 507 nm) and EYFP (emission peak: 527 nm). As the next step, we are planning to develop a two-channel observation system, which will also be capable of observing red fluorescent proteins, such as mRFP (emission peak: 607 nm) and mCherry (emission peak: 610 nm).

#### Excitation system

The point spread function (PSF) of a confocal microscope is expressed by multiplying the distribution of the focused spot of the excitation light by the PSF of the imaging system of fluorescence. In the excitation system, the pinhole is illuminated by the LED light (center wavelength: 470 nm), and the light is diffracted by the pinhole, causing the light to spread at the tube lens side (*Figure 3—figure supplement 1A*). The diffracted light is collected and collimated by the tube lens and focused into the sample by the objective lens. The diffraction of light through a circular aperture is expressed by a well-known formula involving the Bessel function,

$$U\left(\theta\right) \propto \frac{J_1\left(ka\sin\theta\right)}{ka\sin\theta} \tag{A1}$$

where $J_1(u)$, $k$, $a$, $\theta$ denote the Bessel function of the first kind with the order of 1, wavenumber, aperture radius, and diffraction angle, respectively. The angular dependence of intensity can be obtained by taking a square of *Equation 1*; *Figure 3—figure supplement 1B*. The pinhole size in the current setup is 6 μm, at which the diffraction pattern has the first dark ring at about 0.1 radian (*Figure 3—figure supplement 1B*). Since our tube lens has an NA of 0.125, the diffracted light does not fill the entire NA of the lens, and as a result, the effective NA of the focused light into the sample is lower than 0.25 (*Figure 3—figure supplement 1A*). This differs from a typical, laser-based, single-focus scanning confocal microscope. On the other hand, the current configuration is an appropriate choice in terms of light efficiency since most of the diffracted light enters within the NA of the tube lens.

#### Fluorescence imaging system

As for the fluorescence imaging system (*Figure 3—figure supplement 1C*), fluorescence light emitted from fluorescent molecules in the sample is collected by the objective lens with 0.25 NA. Since the fluorescence is emitted isotropically, it fills the entire object-side NA and is focused onto the pinhole by the tube lens with 0.125 NA. If the pinhole diameter is small enough (mathematically represented by the delta function), the focal light field spot focused by the objective lens at the fluorescence wavelength can be considered to correspond to the PSF of the imaging system. *Figure 3—figure supplement 1D* shows the ideal PSF. This PSF was obtained by numerical calculation of the scalar wave integration.

#### Selection of the pinhole diameter

Here, the selection of the pinhole size (diameter) for our optical system is discussed. The pinhole diameter of a confocal microscope is usually set based on the Airy disk diameter of the spot of fluorescence focused on the pinhole. In general-purpose instruments, the pinhole diameter is set to about the Airy disk diameter, but when high spatial resolution is required, it is set to a value smaller than the disk, *e.g.*, half the diameter of Airy disk. At the green wavelength region (*e.g.*, EGFP, EYFP) and the red wavelength region (*e.g.*, mRFP, mCherry), which are currently considered as targets for AMATERAS observation, the Airy disk diameters of the spots focused by our tube lenses (NA = 0.125) are about 5 μm (EGFP: 4.95 μm, EYFP: 5.14 μm) and 6 μm (mRFP: 5.92 μm, mCherry: 5.95 μm), respectively, where we used the following equation for the Airy disk diameter, where $\lambda$ denotes the wavelength.

$$d_{Airy} = 1.22\frac{\lambda}{NA} \tag{A2}$$

The design of the pinhole diameter was considered for both of these systems, excitation and fluorescence imaging. In our optical system, the transmittance of the pinhole disk is low, so it is not desirable to make it smaller than the Airy disk to further reduce its efficiency. On the other hand, if the pinhole is made too much larger than the Airy disk, it is meaningless because it will not provide 3D resolution. Considering that it can be used in both of the two wavelength ranges, we decided on a pinhole size of 6 μm, which is comparable to the Airy disk diameter in the red wavelength range.

## Theoretical and actual resolution in the longitudinal direction

The theoretical resolution (FWHM) in the z-direction in the ideal confocal fluorescence microscope optical system (excitation light is focused to the diffraction limit and detected by a detector of infinitesimal size) is approximately expressed by the following equation with NA, the wavelength of excitation light ($\lambda_{ex}$) and refractive index of immersion medium ($n$).

$$\Delta z_{cf} = 0.64\frac{\lambda_{ex}}{n - \sqrt{n^2 - NA^2}} \tag{A3}$$

The value of $\Delta z_{cf}$ for NA = 0.25, $\lambda_{ex}$ex 470, and n=1 is 9.5 μm, which is considered the best possible resolution that can be achieved with our lens system. As mentioned above, the focusing NA of the excitation light is smaller than the NA of the objective lens, and the pinhole size is slightly larger than the Airy disk. These factors limit the resolution to more than 13 μm at present (*Figure 3D–E*). The challenge for the future is to resolve these issues and approach the ideal resolution.

## Theoretical and actual resolution in the transverse direction

The theoretical spatial resolution (FWHM) in the transverse direction in the ideal confocal fluorescence microscope system is approximately expressed by the following equation.

$$\Delta x_{cf} = 0.37\frac{\lambda_{ex}}{NA} \tag{A4}$$

The value of $\Delta x_{cf}$ for NA = 0.25 and $\lambda_{ex}$ex 470 is 0.70 μm. However, the experimental value (*Figure 3D–E*) is significantly larger than this theoretical value. Similar to the above discussion on the resolution in the z-direction, this is attributed to the fact that our imaging system is far from the ideal system. In addition, the resolution in the transverse direction is also influenced by the secondary relay-lens system that transfers the fluorescence light component passing through the pinhole to the image sensor with magnification of 1×or 2× (*Figure 3B*). Since the NA of the 1×relay lens (0.079) is significantly smaller than that of the tube lens (0.125), large NA component (>0.079) is lost at the edge of the entrance aperture of the relay lens. The effective NA for the fluorescence imaging is 0.158 at the sample space, which is the NA of the relay lens multiplied by the magnification factor (2×). This is the reason why the spatial resolution of AMATERAS-2c in the transverse direction (*Figure 3D*) is downgraded from the wide-field imaging system (AMATERAS-2w) (*Figure 2A*) as well as from the ideal confocal imaging system. The transverse resolution (FWHM) of the theoretical PSF for wide-field fluorescence imaging is expressed by the following equation with NA and emission wavelength ($\lambda_{em}$).

$$\Delta x_{wf} = 0.51\frac{\lambda_{em}}{NA} \tag{A5}$$

The values of $\Delta x_{wf}$ for NA = 0.25 and 0.158 are 1.05 μm and 1.66 μm ($\lambda_{em}$em 515 nm), respectively. This difference explains the degradation of the resultant transverse resolution.

In the case when the 2×relay lens is used, on the other hand, its NA (0.12) is only slightly smaller than that of the tube lens (0.125). The degradation of the transverse resolution is well mitigated compared to the 1×relay lens (*Figure 3E*).

## Note 2. Discussion on the computational sectioning: Parameter adjustment and limitation

### Basic concept

In the optimization of the cutoff frequency of the baseline estimation, we applied the concept of independent component analysis (ICA). As similar to ICA, we assumed that fluorescence image in

the focal plane is away from a Gaussian distribution (normal distribution), but the superposition of images from multiple planes including in-focus and out-of-focus planes brings a distribution closer to a Gaussian distribution. *Figure 4—figure supplement 1* shows how the estimated image varies depending on the cutoff frequency. The raw image (*Figure 4—figure supplement 1A*) is a part of the cardiac organoid shown in *Figure 4* in the main text. The computational sectioning calculation was applied with various cutoff frequencies ($F_c$) from 0.010 to 0.300 $\mu m^{-1}$, and the four types of statistical measures to represent the non-Gaussian nature (non-Gaussianity) were calculated for the estimated in-focus images. The non-Gaussianity measures are kurtosis, skewness, negentropy, and mutual information (*Getreuer, 2013*). The in-focus and out-of-focus images at three cutoff frequencies ($F_c$ = 0.020, 0.063, 0.141 $\mu m^{-1}$) are shown to see how the images change with the frequency (*Figure 4— figure supplement 1B–C*), and the line profiles on the raw image, in-focus, and out-of-focus images are shown in *Figure 4—figure supplement 1D* to clarify their difference.

## Determination of the non-Gaussianity measure and the cutoff frequency

First, we decided which measure we should use for the non-Gaussianity. *Figure 4—figure supplement 1E* shows the cutoff frequency dependence of the four measures, which was calculated for the subregion of image (yellow square region in *Figure 4—figure supplement 1A*). In ICA, maximizing kurtosis or skewness is considered to maximize the non-Gaussianity, whereas minimizing negentropy or mutual information is considered to maximize the non-Gaussianity. Among these measures, negentropy and mutual information have their minimum around 0.141, but the estimated baseline contains the nuclear structure. The in-focus image, obtained by subtracting the baseline image from the original image, is noise-dominant (*Figure 4—figure supplement 1D*, right). This is not suitable for our purposes, so we do not employ these indices. On the other hand, the skewness has a maximum value around 0.063, where the in-focus image appears to have clear nuclei, which is consistent with our objective (*Figure 4—figure supplement 1B*, middle). When the frequency was much lower ($F_c$=0.020) (*Figure 4—figure supplement 1B*, left), the background light component remained in the in-focus image. The kurtosis was found to have a similar frequency response to the skewness, with its maximum value at almost the same frequency. This trend was almost common in other images as well. These discussions led us to adopt skewness (or kurtosis) as non-Gaussianity measures, and set the cutoff frequency to $F_c$=0.063 for this image data, the frequency with the maximum skewness value. This optimization of the cutoff frequency was performed on a subset of a given data prior to the calculation of the entire dataset.

## Limitation of the computational sectioning

Here, we discuss the limitation of applicability of our computational sectioning. As this method involves a baseline estimation process using a low-pass filter with a single cutoff frequency, the structures of the target object should be of uniform size, such as the cellular nucleus. When different sizes are mixed together in the object, the image processing often does not work well. An example of the application of this method to a computer-generated model image is shown in *Figure 4— figure supplement 2*. In the model, a fluorescent sphere with a diameter of 10 $\mu m$ or 30 $\mu m$ is centered as the target object, and a 50 $\mu m$ fluorescent sphere is placed 100 $\mu m$ deep as the source of background light. A pseudo fluorescence image was generated on that 3D fluorescence distribution by the following procedure: (1) A theoretical point-spread function of the wide-field imaging system (NA = 0.25, wavelength = 520 nm) was convolved, (2) the intensity values was adjusted so that the maximum fluorescence intensity was 100, and (3) the values were converted to integers by rounding and Poisson noise was added. *Figure 4—figure supplement 2A* -top shows a fluorescence image of the $z$-plane passing through the center of a 10 $\mu m$ target sphere. The in-focus image was estimated (*Figure 4—figure supplement 2A* -bottom) by the method described above, with the optimal cutoff frequency set to the frequency at which the skewness is maximum, $F_c$ = 0.045 (*Figure 4—figure supplement 2C*). Similarly, *Figure 4—figure supplement 2B* shows the original image of a 30 $\mu m$ sphere and the estimated in-focus image. For the 30 $\mu m$ sphere, a cutoff frequency of 0.022 was found to be optimal (*Figure 4—figure supplement 2C*), with which the in-focus image was estimated (*Figure 4—figure supplement 2B*, bottom left). However, an artifact occurred where the area outside the boundary formed a valley, which happened because the spatial frequencies of the background light and the target sphere were close, leading to insufficient separation. If the cutoff frequency chosen for the 10 $\mu m$ sphere is used for the 30 $\mu m$ sphere image, the image of the target sphere falls within the baseline (*Figure 4—figure supplement 2B*, bottom right). *Figure 4— figure supplement 2D* shows images estimated using several frequencies in the case of 10 $\mu m$ and 30 $\mu m$ spheres side by side. At $F_c$ = 0.045, the optimal frequency for 10 $\mu m$ sphere (*Figure 4—figure*

*supplement 2C*), the 10 μm sphere is still well separated from the background, but the 30 μm sphere is included in the background light. Many artifacts occur at other frequencies as well. These results indicate that it is undesirable to apply this method to fluorescent images of two objects of very different sizes or of objects that are close in size to the background light.

