## [Editor Report · eLife Assessment]

The **important** study established a large-scale objective and integrated multiple optical microscopy systems to demonstrate their potential for long-term imaging of the developmental process. The **convincing** imaging data cover a wide range of biological applications, such as organoids, mouse brains, and quail embryos, but enhancing image quality can further enhance the method's effectiveness. This work will appeal to biologists and imaging technologists focused on long-term imaging of large fields.

---

## [Referee Report · Reviewer #1 (Public review)]

Summary:

The authors are trying to develop a microscopy system that generates data output exceeding the previous systems based on huge objectives.

Strengths:

They have accomplished building such a system, with a field of view of 1.5x1.0 cm2 and a resolution of up to 1.2 um. They have also demonstrated their system performance on samples such as organoids, brain sections, and embryos.

Weaknesses:

To be used as a volumetric imaging technique, the authors only showcase the implementation of multi-focal confocal sectioning. On the other hand, most of the real biological samples were acquired under the wide-field illumination, and processed with so-called computational sectioning. Despite the claim that it improves the contrast, sometimes I felt that the images were oversharpened and the quantitative nature of these fluorescence images may be perturbed.

---

## [Referee Report · Reviewer #2 (Public review)]

Summary:

This manuscript introduced a volumetric trans-scale imaging system with an ultra-large field-of-view (FOV) that enables simultaneous observation of millions of cellular dynamics in centimeter-wide 3D tissues and embryos. In term of technique, this paper is just a minor improvement of the authors' previous work, which is a fluorescence imaging system working at visible wavelength region (https://www.nature.com/articles/s41598-021-95930-7).

Strengths:

In this study, the authors enhanced the system's resolution and sensitivity by increasing the numerical aperture (NA) of the lens. Furthermore, they achieved volumetric imaging by integrating optical sectioning and computational sectioning. This study encompasses a broad range of biological applications, including imaging and analysis on organoids, mouse brains, and quail embryos, respectively. Overall, this method is useful and versatile.

Weaknesses:

What is the unique application that only can be done by this high-throughput system remains vague. Meanwhile, there are also several outstanding issues in this paper, such as the lack of technical advances, unclear method details and non-standardized figures.

Comments on revisions:

The revised manuscript has significantly improved in response to the initial review comments, particularly with the detailed additions regarding the objective lens and confocal imaging modes, which enhance the clarity and comprehensibility of the paper. While the structure and arguments are much clearer overall, there are still key issues that need to be addressed, specifically regarding algorithm validation, computational sectioning presentation, and volume imaging rate.

Algorithm Validation:

The validation of the algorithm's accuracy is not sufficiently robust. Reviewer 1's comment is entirely reasonable, and the authors should validate the algorithm's accuracy using well-established methods as ground truth. In the revised version, the authors attempt to demonstrate the fidelity of the algorithm by employing deep learning methods for high-accuracy cell recognition. However, this validation relies solely on comparisons between deep learning results and manual annotation results. The problem lies in the fact that both manual annotations and deep learning outcomes are derived from algorithm-processed data, which fails to prove the authenticity or validity of the data itself. To strengthen the validation, the authors should incorporate independent, gold-standard methods for comparison.

Computational Sectioning:

In the revised manuscript, the authors effectively demonstrate the ability of optical sectioning to improve axial resolution using fluorescent beads, as shown in Fig. S3, which is a strong point. However, the manuscript lacks a direct comparison for computational sectioning and does not provide a clear evaluation of axial resolution before and after applying computational sectioning. While some related information is included in Figs. 5.C and D, the details are insufficient, and intensity profiles are absent. I recommend that the authors include more direct visual demonstrations of computational sectioning, along with comparisons of axial resolution before and after applying computational sectioning. This would better showcase the method's effectiveness.

Volume Imaging Rate:

The manuscript currently omits critical details about the method's volume imaging rate. In the description of the quail embryo imaging experiment, key parameters such as exposure time and imaging speed are missing. Additionally, the manuscript does not discuss the maximum imaging rate supported by the system in confocal mode. The volume imaging rate is an essential factor for biological researchers to evaluate the applicability of the technique. Therefore, this information should be included, ideally in the abstract and introduction. Furthermore, the authors could describe how the volume imaging rate performs under different conditions and discuss its potential applications across various biological research contexts. Including such details would significantly enhance the paper's utility and appeal to the broader research community.

These adjustments will further strengthen the manuscript and address the reviewers' concerns effectively.

---

## [Author Response]

The following is the authors’ response to the original reviews.

**Reviewer #1 (Public Review):**
Summary:The authors are trying to develop a microscopy system that generates data output exceeding the previous systems based on huge objectives.Strengths:They have accomplished building such a system, with a field of view of 1.5x1.0 cm2 and a resolution of up to 1.2 um. They have also demonstrated their system performance on samples such as organoids, brain sections, and embryos.Weaknesses:To be used as a volumetric imaging technique, the authors only showcase the implementation of multi-focal confocal sectioning. On the other hand, most of the real biological samples were acquired under wide-field illumination, and processed with so-called computational sectioning. Despite the claim that it improves the contrast, sometimes I felt that the images were oversharpened and the quantitative nature of these fluorescence images may be perturbed.
**Reviewer #2 (Public Review):**
Summary:This manuscript introduced a volumetric trans-scale imaging system with an ultra-large field-of-view (FOV) that enables simultaneous observation of millions of cellular dynamics in centimeter-wide 3D tissues and embryos. In terms of technique, this paper is just a minor improvement of the authors' previous work, which is a fluorescence imaging system working at visible wavelength region (https://www.nature.com/articles/s41598-021-95930-7).Strengths:In this study, the authors enhanced the system's resolution and sensitivity by increasing the numerical aperture (NA) of the lens. Furthermore, they achieved volumetric imaging by integrating optical sectioning and computational sectioning. This study encompasses a broad range of biological applications, including imaging and analysis of organoids, mouse brains, and quail embryos, respectively. Overall, this method is useful and versatile.Weaknesses:The unique application that only can be done by this high-throughput system remains vague. Meanwhile, there are also several outstanding issues in this paper, such as the lack of technical advances, unclear method details, and nonstandardized figures.

Here, we address the first part of the Weaknesses concerning the unique application, and will respond to the latter part in the Reply to the Recommendations.

We are developing 'large field of view with cellular resolution' imaging technique, aiming to apply it to the observation of multicellular systems consisting of a large number of cells. Our proposed optical system has achieved optical performance that enables simultaneous observation of more than one million cells in a single field of view. In this paper, we have succeeded in adding three-dimensional imaging capability while maintaining the size of this two-dimensional field of view. By simultaneously observing the dynamics of a large number of cells, we can reveal spatio-temporal sequences in state transitions (pattern formation, pathogenesis, embryogenesis, etc.) in multicellular systems and discover cells that serve as a starting point. These were mentioned in the 1st and 2nd paragraphs of the Introduction section (Line 48-, 58-) and discussed in the 4th paragraph of Discussion section (Line 398-) of the main text. While our previous work on two-dimensional specimens has shown its validity, the present work demonstrated that temporal changes of multicellular systems in three-dimensional specimens can be observed at the single-cell level.

Ideally, we aim to achieve the same level of depth observation capability as the FOV size in the lateral direction. However, at present, the penetration depth for living specimens is limited to a few hundred micrometers due to non-transparency, while the lateral FOV size exceeds 1 cm. The current optical performance is well-suited for systems where development occurs within a thin volume but a large area, such as the quail embryo presented in this paper (Fig. 6 in the revised manuscript). In addition to quail embryos, this technique can also be applied to the developmental systems of highly transparent model organisms, such as zebrafish. Furthermore, for chemically cleared specimens, even those thicker than 1.5 mm, as shown in this paper (Fig. 5 in the revised manuscript), can be observed. Besides organs other than the brain, it could also be applied to imaging entire living organisms. However, for observation depths up to 10 mm, such as in the whole mouse brain, a mechanism to compensate for spherical aberration is required, which we consider the next step in our technological development.

**Recommendations for the authors:**

**Reviewer #1 (Recommendations For The Authors):**
(1) I suggest that authors shall re-examine the quantitative nature of their image processing algorithm. Also, I wonder whether there are parameters that could be adjusted, as images in Figure 3D and 4E seem to be oversharpened with potential loss of information.

As the reviewer pointed out, we recognized that there was an insufficient explanation of the image processing.

Therefore, descriptions on the quantitative nature and parameter adjustments have been added to the text (Materials and Methods, Line 552) and the Supplementary File (Fig. S4-5, Note 2), and these have been referenced in the main text. A summary is given below.

The adjustable parameters in our method include the cutoff frequency of the smoothing filter used in the background light estimation. If the cutoff frequency is too high, the focal plane component will be included in the “background”; if it is too low, background light will remain in the focal plane. The cutoff frequency needs to be optimized within this range. In this optimization, neither the size of the cell itself nor the performance of the optical system was considered; instead, we utilized the concept of independent component analysis (ICA). This approach is taken because the size and structure of cells vary from sample to sample, and the optical properties also vary with wavelength and location, making it impractical to consider each factor for every case. ICA employs a blind separation method, which is based on the principle that individual signals deviate from the normal (Gaussian) distribution, while the superimposition of signals tends to bring the distribution closer to the Gaussian distribution. Several indices have been proposed to quantify the non-Gaussian nature of the distribution, including kurtosis, skewness, negentropy, and mutual information. Among these measures, we empirically found skewness to be the most suitable and robust, and therefore adopted it for our algorithm. The optimal parameters were selected using a subset of the data before applying the calculations of the entire dataset. The determined values were then applied to the entire dataset.

Regarding the oversharpening, we believe that it rarely occurs in the image data shown in the manuscript. In a case where low-frequency structures and high-frequency structures are mixed in the focal plane, oversharpeninglike effect can occur because of the disappearance of low-frequency structures, which is discussed in Supplementary File (Note 2, Figs. S5D). However, in the case of a sample with nearly uniform spatial frequency, such as the nucleus observed in this study, oversharpening is unlikely to occur by setting appropriate parameters as described above. If it appears that some images are oversharpened in the figures, it is due to the contrast of the image.

(2) On the other hand, I am curious how a wide-field fluorescence system may reliably extract information from a denselylabeled sample within axial volume of 200 um, as they showed in the mouse brain in Figure 4. Thus I am skeptical regarding the fidelity and completeness of the signals and cells recorded there. It would be ideal if the authors could benchmark their system performance with a two-photon microscope system, which serves as the ground truth.

The reviewer's suggestion is reasonable; however, we are unfortunately unable to observe the same sample using a two-photon microscope. Instead, we will explain these differences from a theoretical perspective. Two-photon microscopes used for brain imaging typically employ objective lenses with a numerical aperture (NA) of at least 0.5, allowing for 3D imaging with depth resolution ranging from several micrometers down to sub-micrometer levels. In contrast, our method uses a lens system with NA of 0.25, and the optical configuration (focusing NA, pinhole size) are not optimized for resolution (Note 2 in Supplementary File), thus the longitudinal resolution (FWHM) is about 14 microns (Fig. 3E in the revised manuscript). This difference is significant in the brain imaging, where our method may not fully separate all cells in close proximity along the depth axis, as shown in the bottom panels (*xz*-plane) of Fig. 5F of the revised manuscript. Nevertheless, we believe that cell nuclei can be accurately detected in this 3D image using appropriate cell detection methods based on deep learning. To support this claim, we conducted cell detection using the state-of-the-art cell detection platform ELEPHANT and incorporated the results into Fig. 5 (Fig. 5G-I). This figure demonstrates that even with the current spatial resolution, accurate detection of cell nuclei is achievable.

We accordingly added one paragraph (Line 285) in the main text to explain the cell detection method and discuss the results. We also added one section into Materials and Methods for more detail of the cell detection (Line 650).

In conjunction with the revision, the developer of ELEPHANT (K. Sugawara) has been included as a co-author.

**Reviewer #2 (Recommendations For The Authors):**
In my opinion, the following concerns need to be addressed.Major comments:(1) The proposed system's crucial element involves the development of a giant lens system with a numerical aperture (NA) of 0.25. However, a comprehensive introduction and explanation of this significant giant lens system are missing from the manuscript. I strongly suggest that the authors supplement the relevant content to provide a clearer understanding of this integral component.

A detailed description of the giant lens system has been added to the main text (Optical Configuration and Performance, Line 83) and the Materials and Methods section (Wide -field imaging system (AMATERAS-2w) configuration, Line 446). A diagram of the lens configuration has also been included in Fig. 1A. In conjunction with these additions, two engineers from SIGMAKOKI CO. LTD., who made significant contributions to the design and manufacturing of the lens system, have been included as co-authors.

(2) The manuscript introduces a computational sectioning technique, based on iteratively filtering technology. However, the accuracy of this algorithm is not sufficiently validated.

It is challenging to discuss accuracy of the processing results compared to the ground truth, because the ground truth is unknown for any of the experiments. Instead, in the Supplementary File (Notes 2, Figures S4-5), we show how the processing results for the measured and simulated data vary with the parameter (cutoff frequency), illustrating the characteristics of our method. The results suggest that by optimally pre-selecting the parameter, it is possible to successfully separate the in-focus and out-of-focus components. This discussion is related to our response to the first recommendation made by the reviewer #1. Please review our response to Reviewer #1 regarding parameter optimization and oversharpening. Here, as an addition, we describe a discussion of the conditions that must be met in order to perform the calculation correctly, as described below (also included in Note 2, Limitation of the computational sectioning).

To apply this method, certain requirements must be met regarding cutoff spatial frequency and intensity. Regarding cutoff spatial frequency, the algorithm utilizes a low-pass filter with a single cutoff frequency, which can make it challenging to accurately extract structures in the focal plane when structures of varying sizes and shapes are mixed within the sample. This is illustrated by the simulation shown in Fig. S5 and described in Note 2. Conversely, regarding intensity, if the structure’s intensity in the focal plane is weak compared to the Gaussian fluctuations in the background intensity, it becomes difficult to extract the structure. However, intensity fluctuations can be reduced by applying a 3x3 moving average filter to the entire image as a pre-processing step before applying the baseline estimation algorithm.

In the experimental data presented in this paper (Figs. 4-6 in the revised manuscript), the spatial frequency issue was not significant because the target structures, which are stained nuclei, appear to be of nearly uniform size in the focal plane. The second issue, related to intensity, is also addressed in Fig. 4, as the signal intensity from the focal plane is sufficient to overcome background light in almost all regions. In the mouse brain example, the use of confocal imaging suppresses background light, allowing the structures in the focal plane to be accurately extracted.

(3) I didn't see a detailed description of the confocal imaging in the manuscript. If it adheres to conventional confocal technology, then the question arises: what truly constitutes the novel aspect of this technique?

The principle of confocal imaging and optics is based on the use of a pinhole array, a system also employed commercially by CrestOptics (X-Light, Italy). Prior to the 1990s, when the configuration utilizing Yokogawa Electric's pinhole array and microlens array pairs became popular, pinhole array-only setups were the norm, and are now considered somewhat traditional. We do not claim novelty in the optical configuration itself, but rather in the design of a confocal optical system tailored for our original large-field (low-magnification) imaging system with a relatively high NA. The pinhole array disk we designed features significantly smaller pinholes and correspondingly tighter pinhole spacing than those used for high-magnification observation purposes. We believe that this unique size and arrangement provides sufficient novelty.

We have revised the manuscript to clearly emphasize what we believe constitutes the novelty of this technique (paragraphs starting from Line 166 and Line 183). We have also added a discussion on our confocal optical configuration and its spatial resolution in the Supplementary File (Note 1, Fig. S2-3).

(4) Light-sheet and light-field microscopy, as two emerging 3D microscopy techniques which has theoretically higher throughput than confocal, are not sufficiently introduced in this manuscript.

In the previous version, we briefly mentioned light-sheet and light-field microscopy, but we recognized that more detailed explanations were necessary and should be included in the manuscript. We have added several sentences to the main text (Line 159-165). A summary is provided below.

Light-sheet microscopy requires the illumination light to propagate over long distances within the specimen, and many applications necessitate the use of transparency-enhanced tissue. Even when the sample is highly transparent, no existing technique can form thin optical sections as long as 1 cm. Therefore, light-sheet microscopy is not an effective method for the thin, wide, three-dimensional objects that are the focus of this project. Regarding light-field microscopy, it features a trade-off where the number of pixels in the two-dimensional plane is reduced in exchange for the ability to record three-dimensional fluorescence distribution information in a single shot. In our imaging system, the pixel spacing is set to be comparable to the Nyquist Frequency to observe as many cells as possible, meaning that no more additional pixels can be sacrificed. Therefore, the light-field microscopy technique is not suitable for our imaging system.

(5) The fluorescence images of cardiomyocytes derived from human induced pluripotent stem cells (hiPSCs) stained with Rhodamine phalloidin, as presented in Figure 1(E), exhibit suboptimal quality. This may hinder the effective use of the image for biological research. It is imperative that the authors address and explain this aspect, shedding light on the limitations and potential implications of the research findings.

We acknowledge the reviewer’s concern regarding the suboptimal quality of the fluorescence image. Upon further examination, we recognized that the resolution and clarity of the image could potentially limit its utility for detailed biological analysis. To address this, we have re-examined the image size and quality to enhance its presentation in Fig. 2C-E in the revised manuscript, which allows for finer structures to be recognized within the large image size.

Regarding the effective use of the image for biological research, the results shown in the images indicated the capability of observing subcellular structures, such as myofibrils, in cell sheets with a large area, such as myocardial sheets. This would enable us to simultaneously investigate micro-level structures (orientation and density of myofibrils) and macro-level multicellular dynamics (performance of myocardial sheet). We added the above explanation in the manuscript (Line 146). We hope this revision clarifies the quality and utility of the presented image.

(6) The imaging quality difference between the two techniques shown in Figure 1F, G is relatively small, and the signal distribution difference shown in Figure H is significant, unlike the effects expected from an improvement in resolution.

We acknowledge the reviewer's concern regarding the minimal apparent difference in imaging quality between the two images. Upon re-evaluation, we recognized that the original presentation may not have clearly demonstrated the improvements intended by the different techniques. Figure 1H, which showed the line profile of Figs. 1F and G, may have been impacted by the resolution and compression settings of the image file, leading to a less pronounced distinction between the two techniques. To address this, we have enlarged Figs 1F and 1G

(renumbered as Fig. 2D and 2E in the revised manuscript) and carefully reviewed the resolution and compression ratio to ensure that the differences are more clearly visible.

(7) The chart in Figure 2(C) lacks axis titles and numerical labels, making it challenging for readers to comprehend. To enhance reader convenience, it is recommended that the authors incorporate axis titles and numerical labels, providing a clearer context for interpreting the chart.

We appreciate the reviewer’s observation regarding the lack of axis titles and numerical labels in the figure. The vertical axis represents fluorescence intensity, which we initially omitted, assuming it was self-evident. However, as the reviewer correctly pointed out, it is crucial to ensure that figures are clear and accessible to readers from diverse backgrounds. In response, we have added the vertical axis title to Fig. 2C (renumbered as Fig. 3C in the revised manuscript) to enhance clarity, while the numerical labels remain omitted as the unit is arbitrary (a.u.). We have also reviewed all other figures in the manuscript to ensure that no similar errors are present.

(8) In Figures 2(D) and (E), where the authors present the point spread function for quantifying the lateral and axial resolution of the system, I would recommend increasing the number of fluorescent microspheres to more than 10 for statistical averaging. This adjustment would strengthen the persuasiveness of the data and contribute to a more robust analysis.

We appreciate the reviewer’s recommendation to increase the number of fluorescent microspheres for statistical averaging in Figs. 2D and E (renumbered as Fig. 3D-E in the revised manuscript). In response, we have revised the graphs to present the point spread function with the statistical mean and standard deviation (SD) of fluorescent images obtained from a large sample size (N = 100), and accordingly revised the main text to mention the statistics (Line 118, Line 132). We also recognized that a similar adjustment was necessary for Figs 1C and D (renumbered as Fig. 2A-B in the revised manuscript), and have accordingly made the same modifications to those figures as well. We believe these changes enhance the robustness and persuasiveness of our data.

(9) Figure 4(C) visually represents the characteristic 3D structures of several regions. However, discerning the 3D structural information in the images poses a challenge. To address this issue, I recommend that the authors optimize the 3D visualization to improve clarity and facilitate a more effective interpretation of the depicted structures.

We appreciate the reviewer’s suggestion regarding the challenges in discerning the 3D structural information in Fig. 4C. To address this, we have added representative images from the xy-plane and xz-plane of the cortex, medial habenula, and choroid plexus (Fig. 5G-I) in the revised manuscript. These additions provide a clearer visualization of the 3D distribution in each region, making it easier for readers to interpret the structures. Additionally, we have overlaid the results of deep-learning based cell detection on these images, further enhancing the visibility of the cells. This adjustment also aligns with our response to Reviewer #1's second comment.

Minor comments:(1) The labelling of ROI is missing in Figure 1(e).

We appreciate the reviewer’s observation regarding the missing labeling of the ROI in Fig. 1E. Upon review, we confirmed that the ROI was indeed labeled with a white square in the previous manuscript; however, it was difficult to discern due to its small size and the black-and-white contrast. To improve visibility, we have recolored the square in magenta, ensuring that it stands out more clearly in the figure (Fig. 2C in the revised manuscript).

(2) The subfigure order and labeling in Fig. 1 and Fig. 2 are not consistent.

We appreciate the reviewer’s attention to the subfigure order and labeling in Fig. 1 and 2 (Fig. 1-3 in the revised manuscript). To accommodate subfigures of varying sizes without leaving gaps, we arranged the subfigures in a non-sequential order. However, we have ensured that the text refers to the figures in the correct order. We acknowledge the importance of consistency and will work with the editorial team to explore the best way to present the figures while maintaining clarity and alignment with standard practices.

(3) Figure 1B reappears in Figure 2.

We appreciate the reviewer’s observation regarding the repetition of Figure 1B in Figure 2. While the central part of the optical system (custom lens system) is common to both figures, the illumination system, pinhole array disk, and detection optics for the confocal set up differ. To provide a complete understanding of the optical system, we opted to include the full diagram in Fig. 2B (renumbered as Fig. 3B in the revised manuscript). We considered highlighting only the different components, but we felt that doing so might complicate the reader’s comprehension of the overall system. Therefore, we chose to include the common elements twice to ensure clarity.